# Dynamic morphological transformations in soft architected materials via buckling instability encoded heterogeneous magnetization

Neng Xia [1], Dongdong Jin [1,7] ✉, Chengfeng Pan [1], Jiachen Zhang [2], Zhengxin Yang [1], Lin Su[1], Jinsheng Zhao [1], Liu Wang [3] & Li Zhang [1,4,5,6] ✉

The geometric reconfigurations in three-dimensional morphable structures have a wide range of applications in flexible electronic devices and smart systems with unusual mechanical, acoustic, and thermal properties. However, achieving the highly controllable anisotropic transformation and dynamic regulation of architected materials crossing different scales remains challenging. Herein, we develop a magnetic regulation approach that provides an enabling technology to achieve the controllable transformation of morphable structures and unveil their dynamic modulation mechanism as well as potential applications. With buckling instability encoded heterogeneous magnetization profiles inside soft architected materials, spatially and temporally programmed magnetic inputs drive the formation of a variety of anisotropic morphological transformations and dynamic geometric reconfiguration. The introduction of magnetic stimulation could help to predetermine the buckling states of soft architected materials, and enable the formation of definite and controllable buckling states without prolonged magnetic stimulation input. The dynamic modulations can be exploited to build systems with switchable fluidic properties and are demonstrated to achieve capabilities of fluidic manipulation, selective particle trapping, sensitivity-enhanced biomedical analysis, and soft robotics. The work provides new insights to harness the programmable and dynamic morphological transformation of soft architected materials and promises benefits in microfluidics, programmable metamaterials, and biomedical applications.

Three-dimensional (3D) morphable structures with the active transformation capability have attracted great attention due to their geometry-determined functionalities and dynamic interactions with environment[1–7]. Nature organisms often exhibit highly controllable morphological transformations to enhance their adaptability to physical environmental circumstances. For example, a variety of plants with wrinkled surfaces could modulate their hydration by changing surface areas, while glass knifefish ensures the maneuverability and stability of locomotion through regulating its wavy ribbon fin[8–10]. Inspired by the natural behaviors and adaption, researchers developed various soft materials (including crystalline elastomers, hydrogels, and silicone elastomers) and fabrication strategies to achieve the

geometric reconfiguration of 3D structures at different dimension scales[7,11–18]. Studies on the morphological transformation behaviors have significant influence on building intelligent systems with highly tunable properties, designing optimized soft robots, and grasping the underlying mechanisms of biological behaviors[9,19–24].

Burgeoning efforts have been devoted to studying the mechanisms of the transforming structures in quasi-static states and their exotic properties such as cell-alignment[25,26], unusual mechanical, acoustic, and thermal properties in metamaterials[19,27–30]. However, to unlock more complex and diverse transformation modes and systematically unveil their dynamic behaviors, proper design and actuation strategies need to be developed. The transformation processes of soft structures are often driven by the addition and elimination of chemical solvents, and the changing of environmental temperature or pH[1,11,12]. Thanks to the advances in self-driven systems, dynamic regulation of thin-film wavy structures was reported in self-oscillating manners[11,31]. Moreover, anisotropic transformations of the surface-attached structures are exploited through the design of artificial defects or the management of molecular anisotropy[12,32,33]. Although such progress has been achieved, there is still a lack of platforms enabling the architected materials to exhibit highly controllable dynamic regulation and multimodal anisotropic transformations crossing different scales, making it challenging to achieve biomimetic morphological transformation behaviors, unveil their dynamic modulation mechanisms, and explore corresponding physical properties.

In this respect, magnetoactive materials-based soft morphable structures open possibilities to address the limitations via the flexible magnetic field control with excellent controllability and programmability[34–37]. With predesigned magnetization profiles and magnetic field inputs (including magnetic field strength, direction, and gradient), magnetoactive materials have the capability to exhibit rapid and reversible deformation under the stimuli from permanent magnet or electromagnets[7,18,38–44]. Moreover, the dynamic regulation of magnetoactive materials can be achieved by using time-varying controllable magnetic field[43,45–51]. Some previous studies focused on mechanical or metamaterial systems constructed by magneto-elastomers with relatively uniform magnetization profiles, which may limit the degree of deformation freedom and the corresponding functionalities[52,53]. To explore complex transformation of magnetoactive materials, different techniques for achieving programmable magnetization profiles have been developed, including point-by-point programming approach to construct discrete magnetization and template-assisted approach to produce continuous magnetization[7,40,44,54]. However, achieving the buckling, wrinkling or other complex morphological transformation behaviors in magnetoactive materials crossing different scales still remains challenging, mainly due to the difficulty in the construction of complex 3D molds for magnetization or the lack of theoretical models to guide the reverse design of magnetization profile toward desired transformations. A template-free and on-demand manner for the formation of magnetoactive materials with 3D magnetization profile promises benefit in the dynamic geometric reconfiguration of architected materials.

In this work, we present magneto-elastomers-based morphable architected materials with 3D heterogeneous magnetization profiles encoded by buckling instability, which allow the dynamic regulation and multimodal anisotropic morphological transformation. With geometric constraints and absorption of organic solvents (e.g., toluene), the non-freestanding structures could generate diverse buckling transformations, providing a template-free manner for the rapid formation of 3D continuous heterogeneous magnetization crossing different scales. The programmable magnetization technique could be performed with tunable geometric parameters, arrangement of connectivity types and pre-designing artificial defects. By tunning the direction of magnetic stimuli and magnetic field gradient,

reversible multimodal anisotropic transformations of the non-freestanding structures are demonstrated. External magnetic stimulation can help to determine the stable state of soft structure during buckling process, and this regulation does not require prolonged magnetic field stimulation input. With time-varying magnetic field inputs, the dynamic transformations of morphable structures and their impacts on fluidic environments are investigated. Exemplary applications regarding fluidic manipulation, selective particle trapping, sensitivity-enhanced biomedical analysis, and untethered swimming robots are demonstrated. This work provides a wireless actuation platform for the dynamic morphological transformation of architected materials with 3D heterogeneous magnetization profiles, promising benefits in microfluidics and biomedical applications.

## Results and discussion
### Shape-morphing mechanisms of the magneto-elastomers
The 3D morphable structures are fabricated by silicone elastomers doped with ferromagnetic particles (NdFeB particles), as shown in Fig. 1. The magneto-elastomers exhibit swelling characteristic by absorbing diverse organic solvents, e.g., toluene, ethyl acetate, and n-hexane (Supplementary Figs. 1 and 2). These organic solvents diffuse into the elastomeric network and induce the volume expansion of the elastomeric structure[55,56]. In this study, toluene was used for following experiments due to its absorption rate (swelling ratio of magneto-elastomer: 1.58) and relative stability at room temperature. The magneto-elastomer could return to its initial state under ethanol by the rapid diffusion of toluene in ethanol. The fabricated structures are attached to glass substrates with hydrophilic treatment. When the magneto-elastomer is immersed in toluene solution, swelling behavior of elastomer structure is induced by the diffusion of toluene into polymer matrix. While the swelling process of bottom part would be affected by the constraint of substrate, axial compressive forces are formed to provide the buckling condition of the swelled elastomer (the buckled state in Fig. 1). The morphologies of the buckled state could be tuned by the geometrical parameters of the elastomers, the absorption rate of the solvent, the pre-designing of artificial defects, and the arrangement of connectivity types[32].

The geometric transformation of the magnetic elastomers can be controlled by magnetic stimuli. A uniform ~2 T magnetic field produced by a magnetizer is applied to magnetize the magnetic elastomers in the buckled state (remanent magnetic moment: 37.5 emu/g, Supplementary Fig. 3). After returning to the undeformed state, there are anisotropic magnetization profiles in the elastomeric structures. By applying external magnetic stimuli, the surface-attached strip structures and cellular structures composed of multiple strips interlaced together could deform to diverse geometric configurations, which are mainly driven by magnetic torques induced by the misalignment between heterogeneous magnetization profile and external magnetic field. The shape-morphing results could be tuned by different magnetic field inputs, as shown in Fig. 1.

### Characterization of magnetic and solvent responsive behavior of surface-attached structures
The shape-morphing properties of the magneto-elastomers with strip structures actuated by organic solvent and magnetic stimuli are systematically characterized. When immersed in the organic solvent, the top surface of the strip structure deforms into a wave shape. After about 20 min, the deformation of the magneto-elastomers gradually stabilized (Supplementary Fig. 4). The magnetization of the elastomer is performed with a pulse magnetic field and the distributions of magnetic flux density of the strip structure before and after magnetization are measured by a magneto-optical sensor (MagViewS, Matesy, Germany), as shown in Fig. 2a. The magneto-optical result demonstrates a wave-shape distribution of the magnetic flux density along the centerline of strip structure. By applying a time-varying

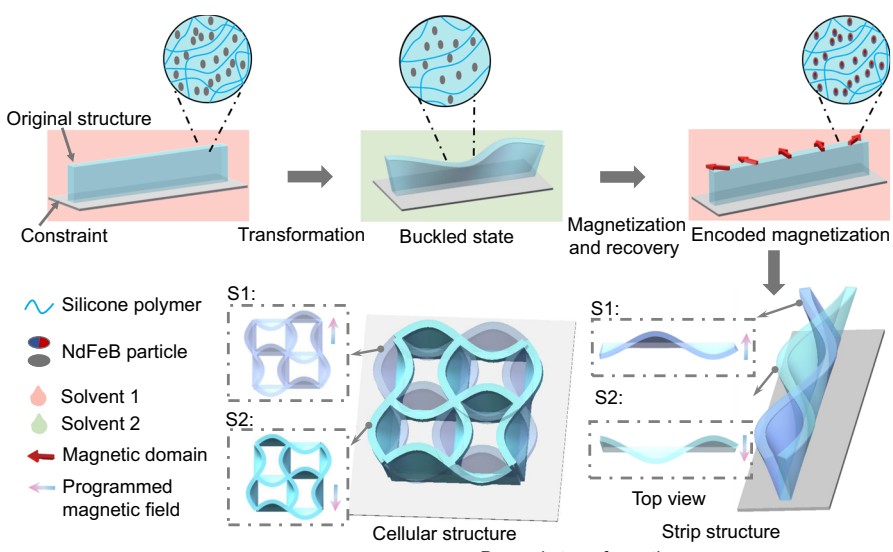

**Fig. 1 | Schematics of dynamic transformation of magneto-elastomers.** The elastomer matrix exhibits reversible deswelling and expansion properties by adding solvent 1 and 2. With geometric constraints and absorption of solvent 2, the elastomeric structure could generate buckling transformation providing a template-free manner for the formation of 3D continuous heterogeneous magnetization. Diverse transformation of elastomeric structures could be achieved by using programmed magnetic inputs.

magnetic field, a 3D wave shape would be formed, and the position of the peaks and troughs changes with the direction of the magnetic field (Fig. 2a and Supplementary Movie 1). The influence of geometrical parameters including length, width, and height of the strip structures on the wavelength and amplitude of the shape-morphing results are characterized by fixed magnetic field strength and magnetic field direction. Under a specific width and height, the length of the non-freestanding structure needs to meet a condition that: $L > L_c$ to form a wavy pattern, where $L$ and $L_c$ represent the length and critical length of the structure, respectively. As shown in Fig. 2b, when $L > L_c$, the wavelength and amplitude of the wavy pattern are not substantially affected by the length of the structure. In addition, the wavelength and amplitude would increase with the increase of the height and thickness of the magneto-elastic structures (Fig. 2c, d). The impacts of geometrical parameters on the deformation results in toluene are shown in Supplementary Fig. 5.

The shape-morphing results are also affected by the magnetic field strength and magnetic field direction. When the direction of the driving magnetic field is consistent with the pulse magnetic field during magnetization, the amplitude of the wavy pattern increases linearly with the increase of the magnetic field strength (Fig. 2e). As shown in Fig. 2f, the shape-morphing results also vary significantly with the height of the structure and the direction angle of the magnetic field. The direction angle represents the angle between the external magnetic field and the direction of the pulse magnetic field during magnetization. When the direction angle is equal to 90∘, the strip structures are twisted as shown in the inset of Fig. 2f with label '1'. When the direction angle is equal to 180∘, the strip structures with the height of 4.3 mm and 5.0 mm collapse without wavy pattern (inset of Fig. 2f with label '2'). While, for the strip structures with the height of 1.6 mm and 2.7 mm, there are still wavy patterns formed in the magneto-elastic structures (inset of Fig. 2f with label '3'). The result demonstrates that when the height of the strip structure is too high, it is easier to collapse under the actuation of magnetic torque. The impact of magnetic field strength and direction angle on the wavelength is shown in Supplementary Fig. 6, where both magnetic field density and direction angle have no effect on the wavelength.

To elucidate the relevant mechanical mechanisms, analytical models are presented for the solvent and magnetic responsive behavior of the strip structures, as shown in Fig. 3 and Supplementary Note 1. The schematic diagram of buckling transformation of a thin elastomer strip is shown in Fig. 3a. The compressive stress would be formed due to the constraint of substrate and the swelling of the elastomer, and when the stress exceeds a critical value, the strip structure transforms into a 3D morphology, which can be described by a sinusoidal function. Foppl-von Karman equations are adopted to derive the quantitative relationship between the buckling results (wavelength and amplitude) and the geometrical parameter, thereby facilitating the forecast of the deformed shape at the free edge of the strip, as shown in Fig. 3c–d. The theoretical model (Eqs. S15 and S18) illustrates that once the critical buckling condition is fulfilled, the length of the strip has negligible effect on the wavelength and amplitude, which are mainly determined by the height of the strip. Moreover, the quantitative description of the 3D buckling pattern of the elastomer allows us to analyze the body rotation of the structure and the formation of heterogeneous magnetization profiles during the magnetization and recovery process, as shown in Fig. 3e, h, i. A strip with the height of 2.7 mm is adopted to validate the theoretical model. The variation of magnetization vector along the $y$ direction at the free ($z = H$) and clamped ($z = 0$) edges is shown in Fig. 3h. The magnetic flux density induced by such heterogeneous magnetization profile is measured by magneto-optical sensor and simulated via FEA software, which show agreement with the experimental result. In addition, a plate theory for thin structure is adopted to unveil the impact of magnetic stimulus on the out-of-plane displacement ($\delta_m$) of the strip. Figure 3i demonstrates the calculated shape at the free edge of the deformed strip is consistent with the experimental result. The theoretical models provide a theoretical guideline for the understanding of the underlying transformation mechanisms and structural design.

## Geometric transformation of magneto-elastomer under the coupling stimulation

In addition to solvent or magnetic stimuli acting separately on the elastomer structure, the effects of coupled magnetic and solvent stimulation on structural deformation with various factors including magnetization, solvent environments, and stimulation sequences are investigated. The magnetization profiles of magneto-elastomer are divided into two types, i.e., buckling instability-encoded

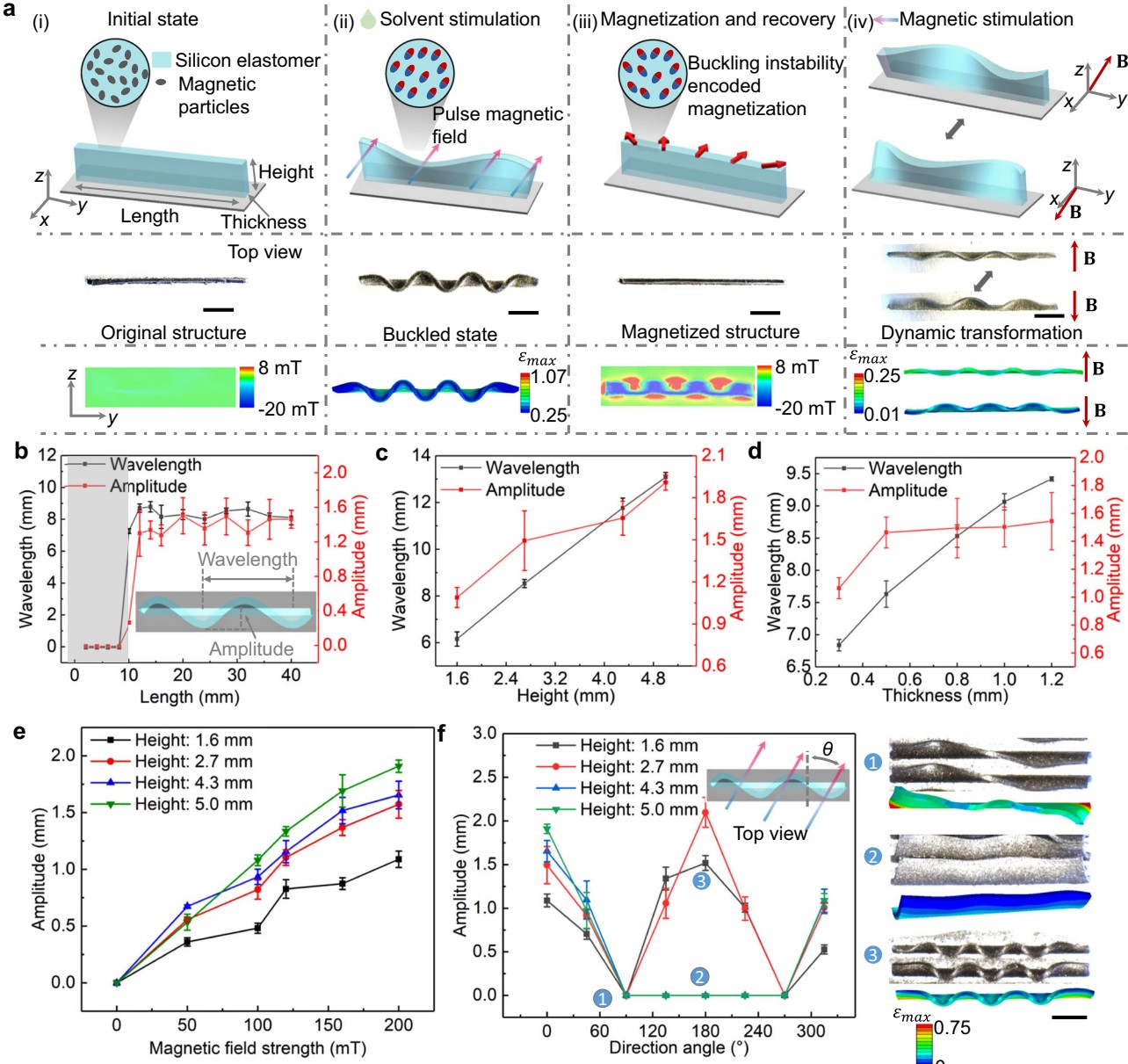

**Fig. 2 | Buckling instability-encoded heterogeneous magnetization in strip magneto-elastomer. a** Demonstration of the original strip structure, its buckled state, and geometric transformation. The magnetic flux density profiles of the structure before and after magnetization are measured by magneto-optical sensor. The modeling results show the deformation of the strip structure under the stimuli from solvent and external magnetic field, and the magnetization profiles of the strip. B represents the applied magnetic field. **b**–**d** The quantitative relationship between wavelength, amplitude of the strip structure and the geometric parameters of the structure including length (**b**), height (**c**), thickness (**d**). The gray area in Fig. 2b represents no wavy pattern generated in the elastomeric structure.

**e**–**f** The quantitative relationship between the magnetic field strength (**e**), direction angle ($\theta$) of magnetic field and amplitude of the buckling structure (**f**). The inset images illustrate experimental and simulation results. The inset image with label '1' shows twisted strip structures when the direction angle is equal to 90°. The inset image with label '2' shows collapsed strip structures with the height of 5.0 mm when the direction angle is equal to 180°. The inset image with label '3' shows the wavy patterns of strip structures with the thicknesses of 1.6 mm when the direction angle is equal to 180°. Error bars stand for the standard error of the mean with the number of trials $n = 3$. $\varepsilon_{max}$ is the maximum principal strain. The scale bar is 5 mm.

magnetization (BM) and uniform magnetization (UM), while the solvent environment includes deionized (DI) water and organic solvent toluene. So, there are 4 coupled stimulus types as shown in Fig. 4a. The buckling instability-encoded magnetization endows the soft structures with more degrees of deformation freedom, as waveforms will be formed following the change of the external magnetic field direction whatever the solvent environment is. In contrast, for the uniformly magnetized structure, by adjusting the direction of magnetic stimulation, the bending peak position will move back and forth within a limited distance along the length direction, leading to the generation

of relatively few deformation modes under the coupled stimulation, as shown in Supplementary Movie 2. To quantitatively describe the morphological changes of the structures upon the coupled stimulation, a phase shift parameter $\varnothing = \frac{\triangle x}{\lambda} \times 360$ is defined and FEA models are developed to verify the buckling modes of the structures, where $\triangle x$ and $\lambda$ represent the displacement of the position of the peak and the wavelength, respectively, as shown in Fig. 4b, c. For BM structures, the phase shift varies with the direction angle of the magnetic field, resulting in a geometrical transformation in the form of traveling waves. The coupled stimulation results illustrate that the magnetic

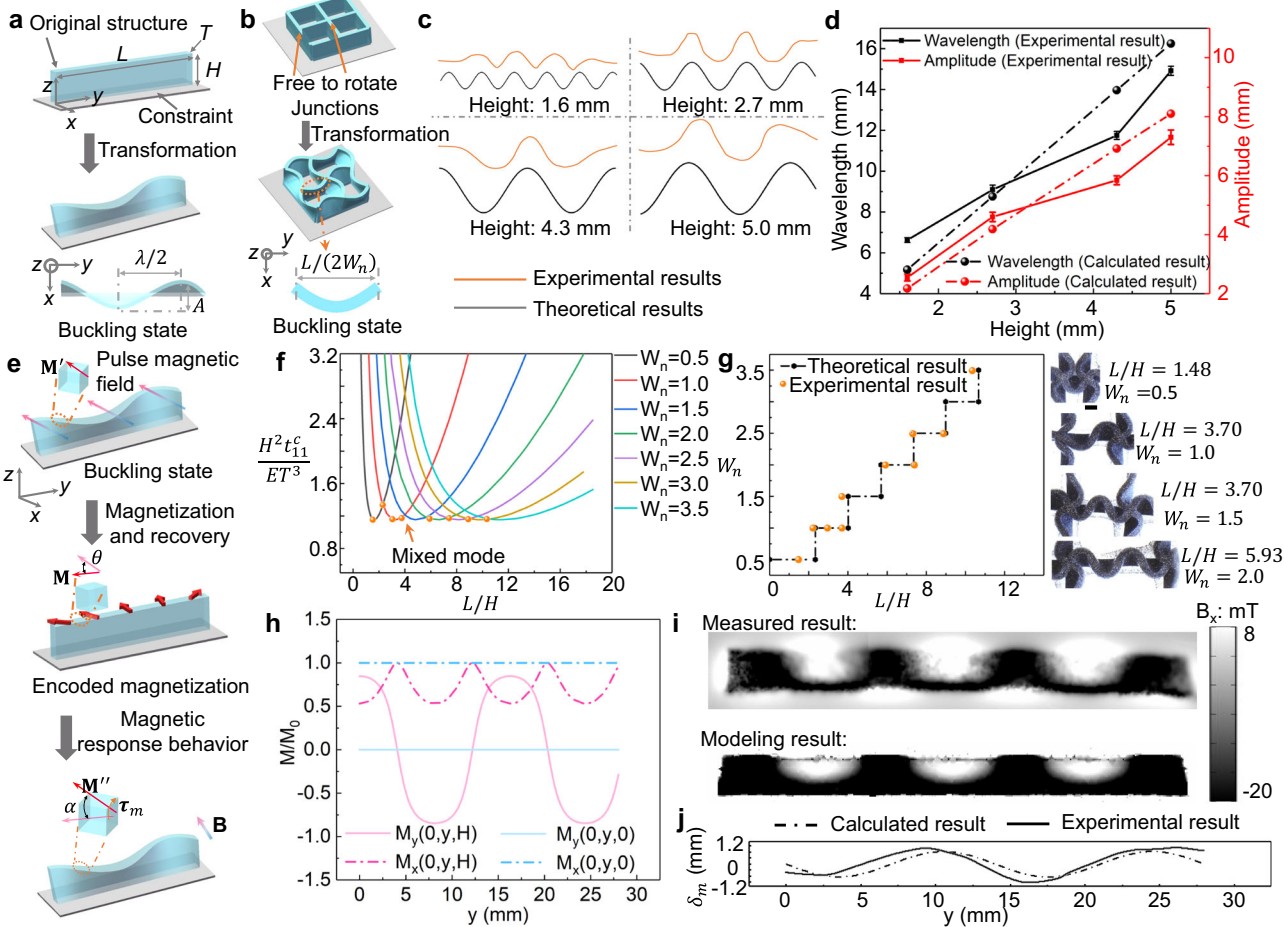

**Fig. 3 | Theoretical model for solvent and magnetic responsive behaviors of soft structures. a** Transformation of elastomer strip with length $L$, height $H$, and thickness $T$ under solvent stimulus. $\lambda$ represents the wavelength. $A$ is the peak-to-peak value. **b** Transformation of elastic plate with free-to-rotate junctions under solvent stimulus. $L$ is the distance between two junctions. The wavelength $\lambda_n$ is determined by $\lambda_n = L/W_n$, where $W_n$ represents the deformation mode of strip. **c** Comparison of the deformed shape of the free edge ($z = H$) of the strip structure with theoretical results. **d** The impact of the height of the strip on the wavelength and amplitude under solvent stimulus. **e** Schematic diagram of the magnetization process, the change of magnetization profile during the recovery process, and the magnetic actuation process. **f** The relationship between the normalized critical force $\left(\frac{H^2 t_{11}^c}{ET^3}\right)$ and aspect ratio ($L/H$) under different deformation modes, where $E$ and

$t_{11}^c$ are Young's modulus and critical membrane force, respectively. Discrete dots in orange color represents the experimental results, which can also be found in Fig. 3g. The intersection of the two curves indicates that mixed deformation modes would occur. **g** Theoretical and experimental results for the relationship between the $W_n$ and the aspect ratio under solvent stimulus. When the aspect ratio reaches a critical value, the strip would exhibit mixed deformation modes. As the experimental results with $L/H = 3.70$, $W_n$ can be equal to 1.0 or 1.5. When $L/H = 7.41$, $W_n$ can be equal to 2.0 or 2.5. **h** Magnetization profile of the strip at $z = 0$ and $z = H$. **i** Validation of magnetic field distribution. **j** Comparison of experimental results and theoretical model for the deformation of the strip under magnetic actuation. The scale bar is 2 mm. Error bars stand for the standard error ($n = 3$).

control strategy enables us to explore controllable buckling deformation of 3D morphable structures, and the buckling instability-encoded magnetization endows the soft structures with more degrees of deformation freedom.

The regulation of buckling states and the impact of stimulation sequences are studied. The buckling modes exhibited by soft structures are multi-stable and the final state after buckling process possesses a certain randomness. As shown in Fig. 4d, compared to the buckling state I induced by toluene alone, the application of coupled stimulation triggers the change to buckling state II with different deformation morphology. Such geometric transformation can be maintained even when the magnetic field is removed. The result indicates that external magnetic stimulation can help to determine the stable state of soft structure during buckling process, and moreover, this regulation does not require prolonged magnetic field stimulation input. The impact of stimulation sequences is studied as shown in Fig. 4e. When a weak magnetic field (20 mT) is applied first followed by the solvent stimulation, the deformation shapes of the elastomeric structure can be dynamically changed by tuning the magnetic field

direction. If the solvent stimulation is applied first followed by magnetic field stimulation, only one deformation shape can be obtained no matter what the field direction. In comparison, when the magnetic field is as strong as 150 mT, it can be found that the stimulation sequences have a negligible effect on the deformation results. Therefore, for the structure in a swollen and buckling state, the input of weak magnetic field cannot provide sufficient energy to overcome the energy barrier for geometric transformation.

## Morphological transformations of cellular structures

The transformations of the non-freestanding structures under different connectivity types are studied. As shown in Fig. 5a and Supplementary Fig. 7, the existence of plates B1-B4 could influence the shape-morphing results of the structure S1 including the position of peaks. According to the distribution of B1-B4, we divide the connectivity types into four types. We set the length of S1 to 8 mm to generate a full waveform under buckled state. Two left-handed structures are formed at both ends of the S1 in Fig. 5a with connectivity type I. For connectivity types I, II, and IV, the two ends of S1 occupy the node position

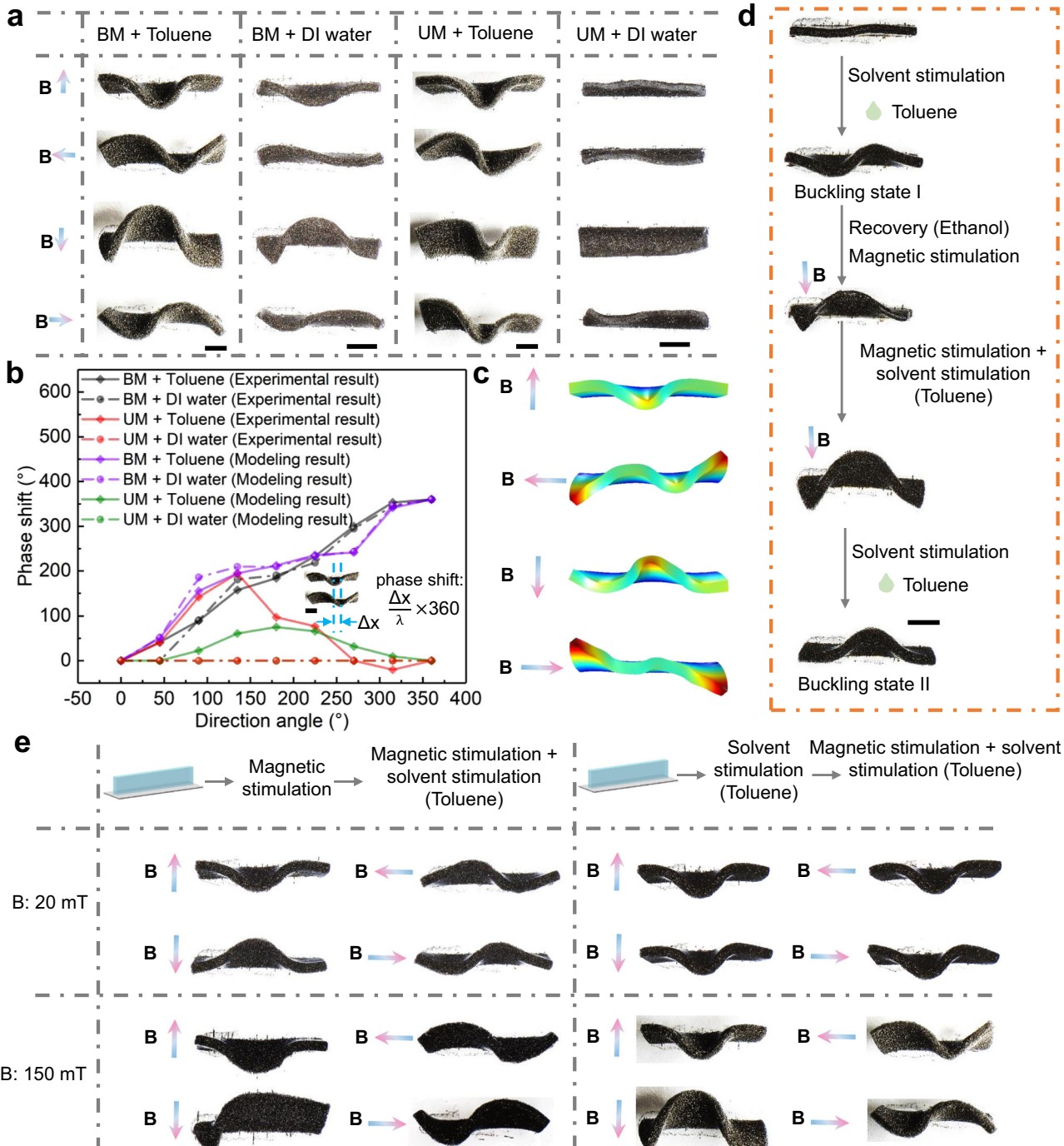

**Fig. 4 | Coupled stimulation for strip structures. a** Impact of different magnetization profiles and solvents on the deformation of strip structure. BM buckling instability-encoded magnetization. UM uniform magnetization. **b** Variation of phase shift with different magnetic field directions and stimuli types. **c** Simulation results for strip with buckling-encoded magnetization profile under the coupling stimulation of magnetic field and organic solvent. **d** Different buckling states of the strip induced by the coupled stimulation. **e** Deformation of strip structures under different stimulation sequences and magnetic field strength. B represents the applied magnetic field. The scale bars are 3 mm.

of the wave structure. While, when the plates are located on one side of the structure S1(connectivity type III), the two ends of S1 form the troughs of the wave structure. The buckling transformations of the non-freestanding structures under different connectivity types are also verified by the simulation results (Fig. 5a). In addition, when the length of S1 is adjusted to 4 mm, the position of peaks in the transformation will change, as shown in Supplementary Fig. 8. For instance, the ends of S1 will occupy the peak position of the weak structure under connectivity type IV. It should be noted that the buckling

transformations for these structures with different connectivity types are also not deterministic, as there will be either up-down or down-up waveforms, as shown in Supplementary Fig. 9. Further study of connections with different constraints of degrees of freedom (including translational and rotational degree of freedom) is performed via FEA models, as shown in Supplementary Fig. 10. The two sides of the plate structure distributed with non-responsive material and fixed constraints would induce a bell-shaped deformation under solvent stimulation, which is different from that with free-to-rotate junctions.

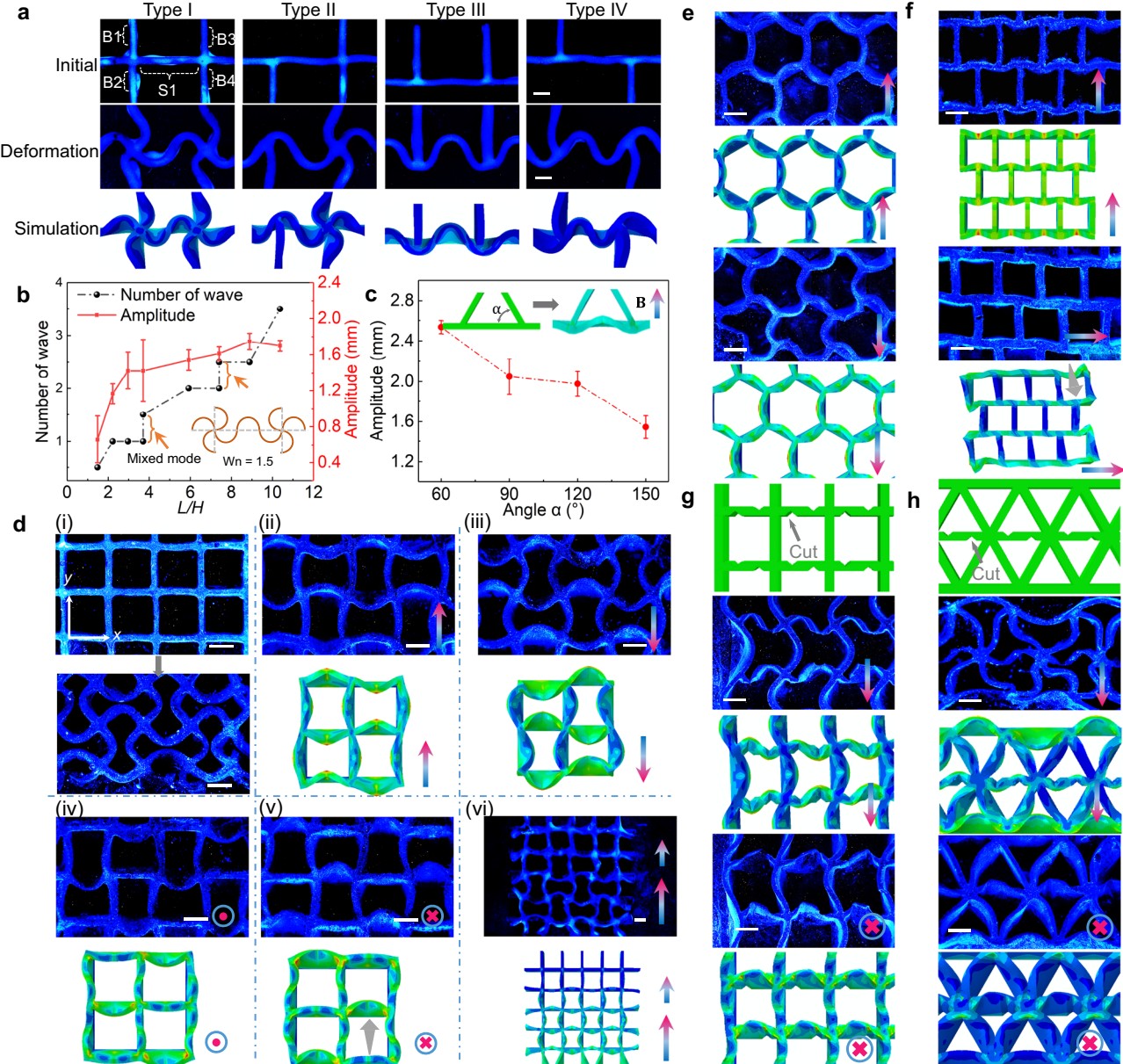

**Fig. 5 | Impact of different connectivity types and geometric transformation of cellular structures. a** Impact of four different boundary types on the buckled configuration in the absorption of solvent. **b** The quantitative relationship between number of wave ($W_n$), amplitude of the buckling structure and the dimensionless geometrical parameter ($L/H$) under magnetic stimulus. $L$ is the distance between the connectivity types, which is the same as the length of S1 in Fig. 5a. $H$ is the height of the plate. $W_n$ represents the number of sinusoidal wave pattern generated in S1 and the inset schematic image shows the state that $W_n = 1.5$. As indicated by the orange arrow, mixed deformation modes are observed. **c** The quantitative relationship between amplitude of the deformation results under magnetic stimulus

and boundary angles ($\alpha$). The inset images illustrate the definition of boundary angle $\alpha$ and exhibit the state that $\alpha = 60°$. **d** Transformations of square lattice structures and simulation results with different magnetic field inputs. The subfigure (i) shows the fabricated square lattice and its buckling configuration in solvent. The subfigures (ii)–(v) illustrate the transformations under different magnetic field directions. (vi) shows the transformation of square lattice with high-field gradient. **e**, **f** Transformations of hexagonal lattice and staggered square lattice. **g**, **h** Transformation of square and triangular lattices with pre-designing artificial defects. The scale bar is 2 mm. Error bars stand for the standard error of the mean with the number of trials $n = 3$.

The deformation results including the number of wave ($W_n$) and amplitude are also affected by the geometrical parameters of the constraints at the two sides. A dimensionless parameter ($L/H$) is adopted, as shown in Fig. 5b. As the $L/H$ increases, the $W_n$ increases in a step-shaped fashion. The theoretical model is also applied to illustrate the impact of $L/H$ on the $W_n$ formed during the solvent stimulation, as shown in Fig. 3b, f, g. With free-to-rotate junctions in Fig. 3b. $W_n = L/\lambda_n$, and $2W_n$ is an integer determining the deformation mode of the soft structure. There is a nonlinear relationship between $L/H$ and normalized critical force $\frac{H^2 t_{11}^c}{ET^3}$ under different deformation modes, where $E$ and

$t_{11}^c$ are Young's modulus and critical membrane force, respectively. The theoretical model indicates that the relationship between $L/H$ and $W_n$ is in a form of step function, which is consistent with the experimental results. When the aspect ratio $L/H$ is within a certain range, the deformation mode of the plate is fixed; when the $L/H$ reaches a critical value, the plate would exhibit mixed deformation modes (Supplementary Fig. 11). For instance, when $L/H = 3.70$ ($L$ and $H$ are 10 and 2.7 mm in the experiments, respectively), the number of wave exhibited by the buckled structure can be 1.0 or 1.5. In addition, the wave amplitude increases with the $L/H$ and gradually flattens out.

Furthermore, adjusting the relative angle between the boundaries would affect the amplitude of the wave, as shown in Fig. 5c.

With the analysis of the transformation under different connectivity types, we extend the deformation to various lattice structures. We first focus on the morphological transformation of a surface-attached square lattice and validate its shape-morphing results using finite element simulations (Fig. 5d and Supplementary Movie 3). A cuboid permanent magnet (length: 40 mm, width: 40 mm; thickness: 25 mm) is used to actuate the lattice structures and the magnetic field distribution generated by the permanent magnet including the magnetic field direction, strength, and gradient is shown in Supplementary Fig. 12. Two distinct transformation modes of square lattice could be achieved by tuning the magnetic field direction. When immersed in toluene solutions, the square lattice exhibits an ordered sinusoidal pattern in which each plate deforms into a half-wavelength structure. First, by applying an in-plane magnetic field parallel to that used for magnetization, the square lattice exhibits a similar pattern to the buckled state in the toluene solution. When the magnetic field direction is rotated by 180 degrees, the lattice structure still forms wavy pattern, but there is a symmetrical relationship with the previous pattern. The transformation results show that the handedness of each node in the lattice structure can be regulated by the magnetic field. Second, the applied out-of-plane magnetic field leads to an anisotropic pattern of the square lattice. The deformation amplitude of the vertical plate is small, while the horizontal plate will deflect in the same direction, and the deflection amplitudes of two adjacent plates in the horizontal direction are different. Third, the area-specific deformation of the square lattice could be achieved by utilizing the gradient of the external magnetic field, which induces a gradual wavy shape along the $y$ direction. In addition, the impact of the coupled stimulation on the deformation of lattice structure is also studied, as shown in Supplementary Figs. 13 and 9, which are consist with the results of aforementioned strip structure, demonstrating the capability of regulating the buckling states of lattice structures.

The responses of architectures with different geometries and cut designs are also investigated. As shown in Fig. 5e and Supplementary Fig. 14, the hexagon lattice could deform into different achiral patterns via magnetic stimuli and the rotations of all vertices in the structure are controllable. In addition, a staggered brick-wall structure, which has been reported to have negative Poisson's ratio property is considered[57–59]. By applying in-plane magnetic fields, the architected material presents a re-entrant hexagonal or a diamond-like pattern (Fig. 5f). Further, the anisotropic transformation of the lattice structures can be tuned by the introduction of predesigned artificial defects. As shown in Fig. 5g, h, two indents are arranged on each side of the plate along the $x$-axis. These indentations would also affect the number of wave and node locations of the transformation of each plate in the lattice structure. For instance, in Fig. 3g, the plates parallel to the $x$-axis and $y$-axis constitute a full wavelength and half the wavelength, respectively. The arrangement of the indentations could enhance the flexibility of the transformation.

## Flow field induced by the morphological transformation under dynamic magnetic field

The capability of promoting fluidic transport is one of the main properties of the dynamic transformations of the magneto-elastic structures. In many living systems such as electric ghost knifefish and cuttlefish, they would transfer momentum with the surrounding fluid environment through the fluctuation of ribbon fins[9,10]. To begin with, we investigate the dynamic flow field induced by the dynamic regulation of non-freestanding strip structures based on the particle image velocimetry (PIV) measurements. We use glycerol with the dynamic viscosity of 0.876 Pa.s as the fluid environment and track the trajectories of the fluorescent particles by performing PIV analysis. We choose a

plate of 12 mm in length that enables the generation of a full waveform along the $y$-axis direction shown in Fig. 6a, b. To achieve its prescribed transformation result in buckled state, an indentation is arranged at the center of the plate, as shown in Supplementary Fig. 15. An in-plane rotating magnetic field generated by a sphere permanent magnet (Supplementary Fig. 16) is applied to induce its dynamic geometric transformation and the impacts of the magnetic field strength and rotating frequency on the flow fluid are studied (Supplementary Fig. 17). The intensity of magnetic field is tuned by the distance between the sphere permanent magnet and the substrate and the variation of magnetic field strength with the distance is shown in Supplementary Fig. 16. With a high-intensity rotating magnetic field (Fig. 6a), the net fluid flow along the $y$-axis is induced around the strip and the direction of the net fluid flow can be adjusted by the rotating direction of the permanent magnet. However, when driven with weaker magnetic field strength and lower frequency, the net fluid flow generated by the strip structure is along the $x$-axis direction. A phase diagram illustrating the fluid flow pattern with different magnetic field strengths and frequencies is shown in Supplementary Fig. 17. The quantitative relationship between the maximum fluid velocity, magnetic field strength, and rotating frequency is shown in Supplementary Fig. 18.

Benefiting from the buckling instability induced magnetization strategies and the pre-designing of artificial defects, we could form strip structures with different magnetization profiles attached to the same substrate, as shown in Fig. 6c and Supplementary Fig. 19. With the arrangement of indentations, the two strips could form mirror-symmetric transformations in buckled states. Under the in-plane rotating magnetic field, the strips could generate more complex net fluid flow (Supplementary Movie 4). The area enclosed by the white dashed box in Fig. 6c represents the region occupied by the strips. During the dynamic transformation of the non-freestanding structures, the fluid flows from the two sides of the white box to its middle area, and then flows out to both sides along the $x$-direction after gathering in the middle area. When the magnets rotate in opposite direction (clockwise direction), the resulting flow field remains consistent (Supplementary Fig. 19). In comparison, when the ribbon array is magnetized in the same buckled state, the fluid transport property generated by its dynamic transformation is the same as the property of a single ribbon (Supplementary Fig. 19). By using surface-attached strip array with the same magnetization state, we could reversibly transport liquid droplet along a rectangular trajectory (Fig. 6d and Supplementary Movie 5). Further, we combine ribbon arrays with mirror-symmetric magnetization and the same magnetization state. In this way, we could use the properties of the flow field generated by the strip structures to merge the two separated droplets and transfer the mixed droplet to the targeted position (Fig. 6e and Supplementary Movie 6). The fluid behaviors of the dynamic regulation of more complex non-freestanding structures including cellular lattice are also investigated (Supplementary Fig. 20). In addition, we calculate the energy consumption of the developed fluidic manipulation method, and perform the comparison of the pumping performance reported by other literatures, as shown in Supplementary Note 2 and Supplementary Table 1, which demonstrates that the developed flow generation method outperforms most reported soft pumping device in terms of pumping efficiency.

## Potential applications of the dynamic transformation of non-freestanding structures

The dynamic modulation of non-freestanding structures provides opportunities for applications in selective particle trapping and sensitivity-enhanced biomedical analysis. Firstly, geometric reconfiguration of cellular surface can be performed for micro-particles sorting and trapping. The processing method is illustrated

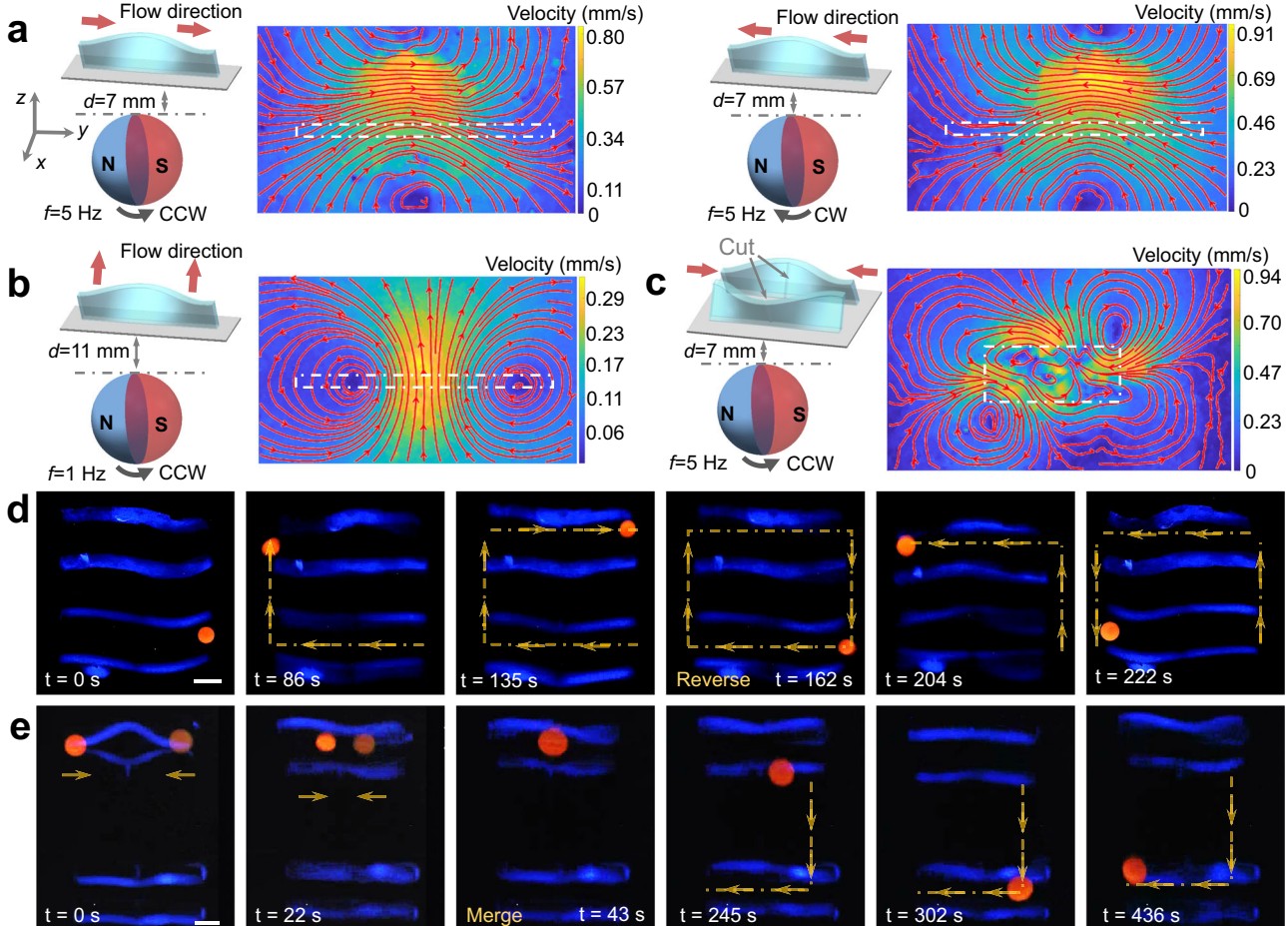

**Fig. 6 | Dynamic flow field generated by the dynamic transformation of the strip structures. a, b** Flow field induced by single strip structure under different magnetic field strength and rotating direction. **c** Flow field induced by two strip structures with symmetric shape-morphing results, which are induced by symmetric cuts arranged in strip structures. **d, e** Demonstration of droplet manipulation with the dynamic flow field generated by the multiple strip structures. **d** The droplet reversibly completes a rectangular trajectory via the trip arrays, which exhibit the same shape-morphing results. **e** The droplets complete series of tasks including merging and transport by using four elastomeric structures, where two of them show symmetric buckling pattern and the other two strip structures show the same shape-morphing results. *d* represents the distance between the substrate and the sphere permanent magnet and *f* represents the rotating frequency of the magnet. The scale bar is 2 mm.

in Fig. 7a. By applying an in-plane rotating magnetic field, the cellular surface could continuously switch between deformed state (each plate in the square lattice deforms into a half sinusoidal wave) and open state. The geometric reconfiguration of cellular surface enables the non-freestanding structure to serve as a miniature vibrating screen to sort particles of different sizes, where based on the chamber formed by the lattice, small particles would be trapped into the closed chamber of the cellular lattice and large particles would be released. To demonstrate the selective trapping function for microparticles, a square lattice whose each plate owns the length of 500 μm is fabricated with a micromold, and three kinds of particles are used with the size of 200, 400, and 600 μm (Supplementary Movie 7). At first, dynamic transformation of cellular structure is triggered by a rotating magnet, resulting in the extrusion of 600 μm particles, and the reservation of 200 and 400 μm particles in the cellular chamber. Afterwards, a static magnetic field is imposed to deform the cellular structure with the formation of a ~340 μm gap, followed by the application of a flow disturbance. At this time, 400 μm particles are physically held by the boundary of cellular structure, while 200 μm particles are flushed away from the deformed gap of chamber. Therefore, this strategy can successfully screen out 400 μm particles from the group of 200, 400 and 600 μm particles. In addition, Supplementary Fig. 21 and Supplementary

Movie 7 show the separation of microparticles by only using the dynamic transformation of the microcellular structure, which allows the reservation of particles smaller than the opening size (500 μm) in chamber. For instance, 200 μm or 400 μm particles can be selectively trapped.

In addition, we demonstrate the collection of aerosol droplets with the flow field induced by the dynamic transformation for highly sensitive biomedical detection. Small droplets generated by spraying or coughing are randomly distributed on a surface and the detection of the respiratory droplets remains a challenging task. Therefore, collecting and detecting aerosol analytes with small droplet size and low concentration are of particular interest toward respiratory related biomedical analysis[60,61]. The flow field induced by the dynamic modulation of the magneto-elastomer-based non-freestanding structures can significantly enrich random sprayed tiny droplets and enhance related detection (Fig. 7b and Supplementary Movie 8). To demonstrate the exemplary application, sprayed droplets doped with fluorescein isothiocyanate, which has been widely used for antibody label and rapid detection of pathogen are adopted[62,63]. At the initial state, the fluorescence signal of the distributed droplets could be barely detected. In Fig. 7b(ii) and (iii), the accumulation of sprayed droplets via the dynamic regulation of non-freestanding strip structures are demonstrated, which

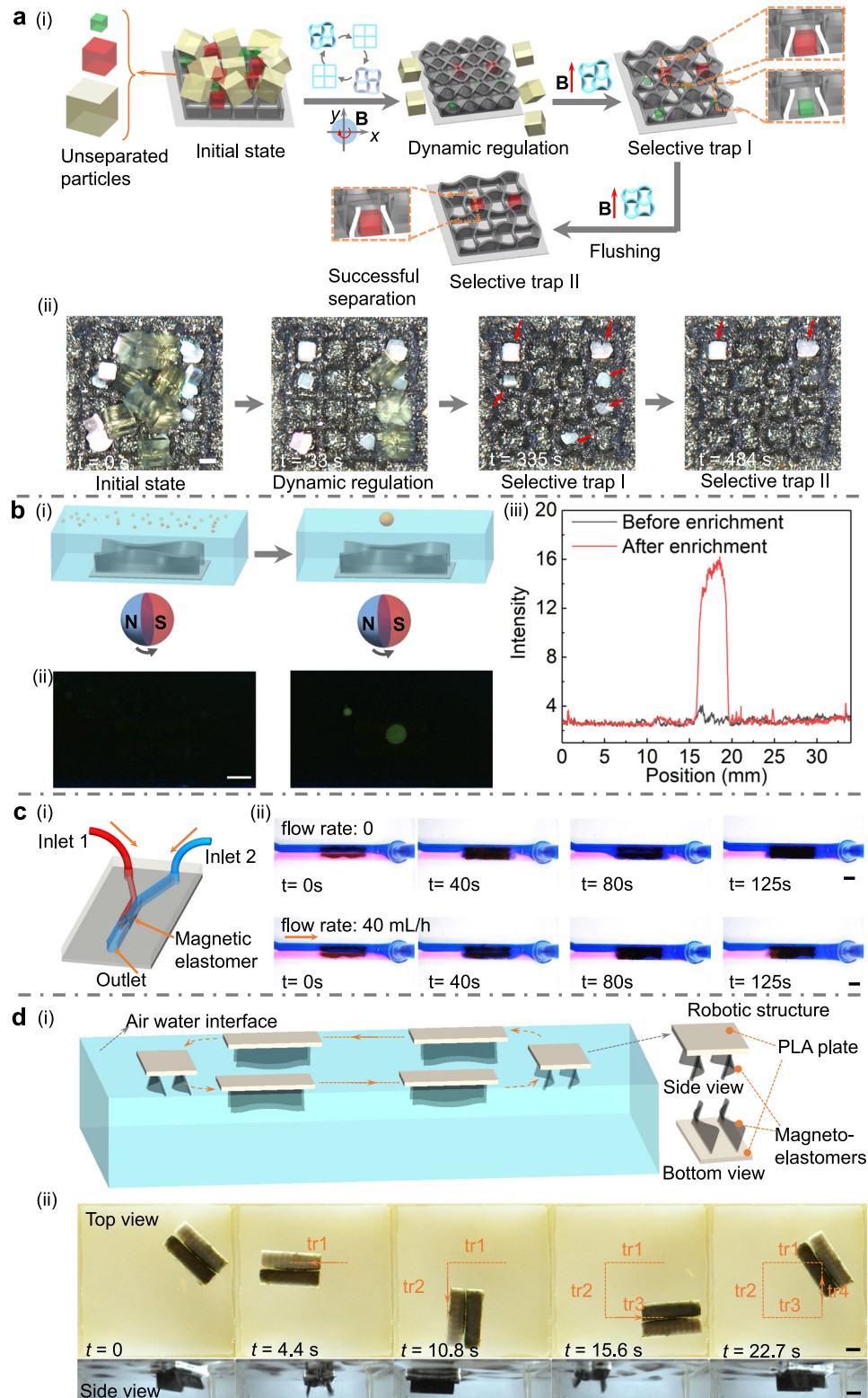

led to an increase in the fluorescence signal that could be easily detected.

The mixing of fluids in small-scale channels is important for the development and application of microfluidic devices. Nevertheless, efficient mixing is challenging due to the low *Re* environment and slow diffusion effects. Existing methods include passive fluid mixing devices, which often require specific channel designs, such as spiral microchannels, or straight channels with groove or ridge microstructures[64]. In addition, active mixing devices have also been developed, including magnetically actuated artificial cilia and bubble-based microfluidic devices, which may have limited mixing thoroughness or need special design of ciliary arrays to achieve optimized metachronal wave[65]. Here, we apply magneto-elastomers with dynamic structural transformations for fluid mixing at low *Re* in microfluidic devices, as shown in Fig. 7c, Supplementary Fig. 22, and Supplementary Movie 9. Glycerol colored with different dyes is used as the fluid at

**Fig. 7 | Exemplary applications of the transformation of the magneto-elastomers. a** Schematic and demonstration of in situ particle manipulation with the dynamic transformation of the cellular structure. **a-i** Schematic illustration of the selective trapping process. 200 (green), 400 (red) and 600 (yellow) μm particles are used here. Dynamic transformation of cellular structure triggered by a rotating magnet results in the extrusion of 600 μm particles. Afterwards, a static magnetic field is imposed to deform the cellular structure with a gap of ~340 μm, followed by the application of a flow disturbance. At this time, 400 μm particles are physically held, while 200 μm particles are flushed away. **a-ii** Experimental results of the selective separation and trapping of particles according to the deformed lattice size. The scale bar is 200 μm. **b** Aerosol droplets collection for sensitivity-enhanced biomedical analysis. **b-i** Schematic of the droplet collection by the dynamic transformation of the magneto-elastomers. **b-ii** Fluorescence images of the sprayed aerosol and collected droplet doping with fluorescein isothiocyanate. The scale bar is 4 mm. **b-iii** Relationship between average intensity and position before and after enrichment. **c** Fluid mixing at low Re using the dynamic transformation of the magneto-elastomers. **c-i** Schematic diagram of the microfluidic device with Y-shaped fluidic channel. The fluid mixing device includes glycerol layers with two colors and magneto-elastomers. **c-ii** Efficient fluid mixing with 0 and 40 mL/h flow rate of the inlet. The scale bars are 3 mm. **d** Untethered swimming robots actuated by the dynamic geometric transformation of magneto-elastomers. **d-i** Schematic diagram of the propulsion of the swimming robot at the air-water interface and the robotic structure. **d-ii** Top view and side view of the swimming robot during the actuation to fulfill a rectangular trajectory. The scale bars are 5 mm.

the inlet and fills the entire microfluidic channel. The volume of the straight channel is 182 mm³, and no apparent diffusion of fluid occurs in the channel under static state. Two magnetic strips with a height of 1.6 mm are used for the fluid mixing. With a dynamic magnetic field, we demonstrate that this mixing device could rapidly mix viscous glycerol with local *Re* less than 0.004. In addition, we also verify the efficient fluid mixing under external dynamic fluid environment (flow rate: 40 mL/h).

Underwater creatures will generate propulsion or achieve movement stability and controllability by controlling wave-like fins, which provides inspiration for the development of biomimetic swimming robots. Ribbon fin-based propulsions have been achieved at the centimeter scale (20–50 cm) by using electromechanical systems to control the locomotion of multiple nodes on the ribbon[66]. But, this strategy is difficult to build smaller-scale untethered swimming robots due to the limited on-board space. In comparison, magnetic control method exhibits high controllability and programmability for small-scale soft robots. By utilizing the proposed buckling instability-encoded heterogeneous magnetization profiles and programmable magnetic field inputs, an untethered swimming robot is developed to achieve the effective swimming behavior with excellent controllability. We adopt a polylactic acid flat plate as the substrate (20 mm in length), and coat a thin layer of silicone elastomer on its surface to facilitate the robust bonding between the substrate and the magneto-elastomers (16 mm in length and 5 mm in height), as shown in Fig. 7d and Supplementary Fig. 23. An oscillating magnetic field generated (Supplementary Note 3) by three-axis Helmholtz coils is applied to actuate the dynamic transformation of the magneto-elastomers and achieve the efficient propulsion of the swimming robot at the air-water interface. The input magnetic field strength and frequency are 9 mT and 12 Hz, respectively. By continuously adjusting the direction angle of the magnetic field, the robot could achieve omnidirectional locomotion and move along a rectangular trajectory, as shown in Fig. 7d and the Supplementary Movie 10. Further increasing the strength of the output magnetic field could improve the magnetic control capability and contribute to the miniaturization of the proposed swimming robot.

In summary, we present magnetic control strategy to achieve dynamic morphological transformation of buckling configurations and exhibit switchable fluidic properties. A facile magnetization programming method by using the buckling instability phenomenon is proposed, which provides a template-free and on-demand manner for the formation of magnetoactive materials with 3D magnetization profile. Through programming magnetic field inputs including magnetic field direction, strength, and gradient, the buckling-induced magnetized elastomers could exhibit multi-mode anisotropic transformations or robust switch between different buckling states. The introduction of magnetic stimulation could help to predetermine the buckling states of soft architected materials, and enable the formation of definite and controllable buckling states without prolonged magnetic stimulation input. The dynamic geometric reconfiguration based on the time-varying magnetic field enables 3D buckling structures to exhibit a series of switchable fluid properties such as directional flow, mixing, and vortex. With the dynamic modulation strategy, we showcase potential applications of the 3D morphable structures in fluidic manipulation, selective object trapping, sensitivity-enhanced biomedical analysis, efficient fluidic mixing in microfluidic device, and untethered swimming robot. In the future, it is envisioned to combine advanced 3D printing technologies such as multi-material 3D printing and 4D printing technologies, to achieve more complex hierarchical geometrical transformation, build customizable 3D dynamic regulation systems, and modify the material preparation process to endow structures with new functions, such as the absorption of environmental pollutants as well as the selective permeation of molecular substances, which promise benefit in the promotion of intelligent cleaning and substance separation technologies.

## Methods
### Materials
NdFeB microparticles with an average size of 38 μm (LW-N-400) were purchased from Guangzhou Xinnuode Co., Ltd. Ecoflex 00-30 was purchased from Smooth-on Inc. Triethoxy(1H,1H,2H,2H-perfluoro-1-octyl)silane (POTS), glycerol, silicone oil (~200 mpa.s), fluorescein isothiocyanate, rhodamine B, methylene blue, crystal violet, isobornyl acrylate were obtained from Aladdin Chemicals. Sylgard 184 were obtained from Dow Chemical Company Ltd. Photoreactive resin PEGDA was obtained from BMF precision, China. All chemicals were used without further purification.

### Preparation of magneto slurry
The magnetic elastomers were prepared by blending silicone-based materials (Ecoflex 00-30 part A and part B) and NdFeB magnetic microparticles in a 1:1 mass ratio. After thoroughly mixing, vacuum degassing was performed to remove air bubbles for 1 min in a vacuum chamber.

### Fabrication of 3D non-freestanding structures
(i) Fabrication of surface-attached strip structures: First, various groove patterns (e.g., rectangular holes) were cut on the acrylic sheets by a laser processing platform (Universal Laser Systems PLS6.75). Hydrophobic treatment was performed for the surfaces of acrylic sheets with groove patterns: Acrylic molds was treated with plasma for 10 min following with silane steam treatment for 3 h using triethoxy(1H,1H,2H,2H-perfluoro-1-octyl)silane (POTS). A glass substrate with hydrophilic surface obtained by plasma treatment for 10 min was prepared. The acrylic mold and glass substrate are held together by clips. The uncured magneto-elastomer was poured onto the surface of the acrylic mold, and the entire structures were placed in a vacuum chamber for 5 min so that the elastomer could thoroughly fill the cavity portion of the acrylic mold. After degassing, the magneto-elastomer was cured in an oven (60 °C) for 20 min. The elastomer structure was peeled off from the acrylic mold and stayed on the glass substrate and the strong adhesion between the elastomer structure and the glass substrate allows the formation of stable non-freestanding structures.

To endow the structure with fluorescent effect, a thin layer of silicone elastomer doped with fluorescence powder (weight ratio 1:0.08) was added on top of the elastomeric structure.

(ii) Fabrication of surface-attached cellular structures: Laser cutting was performed to form lattice-shaped acrylic sheets by the laser processing platform. The lattice-shaped acrylic sheets were placed in a container and PDMS prepolymer (prepared by mixing base and curing agent of sylgard 184 with a weight ratio 10:1) was poured into the container. The PDMS polymer was cured in an oven (60 °C) for 1 h and after that the acrylic sheets were peeled off to form PDMS-based mold with lattice-shaped groove. Hydrophobic treatment was performed for the PDMS-based mold: the PDMS mold was treated with plasma for 10 min following with silane steam treatment for 3 h using triethoxy(1H,1H,2H,2H-perfluoro-1-octyl)silane (POTS). After the silane treatment, the PDMS surface was cleaned via ethanol and dried in the oven (60 °C) for 1 h. The magnetic slurry was poured onto the surface of the PDMS mold, and were placed in a vacuum chamber for 5 min so that the slurry could thoroughly fill the cavity portion. A glass substrate with hydrophilic surface obtained by plasma treatment for 10 min was covered on top of the magnetic slurry, which was cured in the oven (60 °C) for 20 min. The cellular elastomer structure was released from the acrylic mold and stayed on the glass substrate.

(iii) Fabrication of surface-attached microcellular structures: First, a resin mold ($9 \times 9 \times 1.4$ mm$^3$) with lattice-shaped microgrooves was fabricated by a 3D printer (NanoArch S130, BMF Precision, China) and poly(ethylene glycol) diacrylate (PEGDA) resin. To avoid the residual resin affecting the curing of the elastomer, the resin mold was irradiated by UV light for 1 h, and cleaned via ethanol, and followed by the hydrophobic treatment. The magnetic slurry fills the microgrooves of the resin mold in a vacuum chamber and is covered by a hydrophilic-treated glass substrate. After the magnetic elastomer was cured, the cellular microstructure was peeled off from the resin mold.

## PIV analysis
The liquid environment for the PIV analysis was formed by glycerol and fluorescence powders were used as tracer. The distance between the air-water interface and the upper surface of the elastomer structure was kept at 3 mm. Hirox RH-2000 Digital Microscope was used to record the process. The flow field was obtained by performing PIV analysis using an open-source package MatPIV 1.7 (https://www.mn.uio.no/math/english/people/aca/jks/matpiv/).

## Droplet manipulation via the transformation of the strip structures
The ethanol solution in which Rhodamine B was dissolved was used as a fluorescent droplet. As shown in Fig. 6, the fluorescent droplets distributed on the upper layer of the container under UV light irradiation excited red light. The two fluorescent droplets in Fig. 6e have different Rhodamine B contents (mass ratio of 0.02% and 0.075%) to make them exhibit different fluorescence intensities. The viscous liquid environment is formed by silicone oil (~200 mpa.s). The dynamic magnetic field is generated by a rotating sphere magnet with a diameter of 35 mm, as shown in Supplementary Fig. 16, and the surface magnetic field strength of the magnetic sphere is 200 mT. The rotational frequency of the sphere magnet during the manipulation process is 5 Hz. The distance between the upper surface of the sphere magnet and the elastomer structure is about 4 mm. During the actuation process, the locomotion of the droplet is controlled by adjusting the rotation direction and position of the sphere magnet.

## In situ particle manipulation
Particle manipulation was performed underwater. Air bubbles inside the elastomer structure were removed by placing it in a vacuum chamber for 5 min. 200 and 600 μm particles in Fig. 5(b) were fabricated by projection micro stereolithography of PEGDA resin. 200 and

400 μm fluorescent particles in Supplementary Fig. 21 were prepared by 3D printing of photocurable resin composed of PEGDA, isobornyl acrylate, and green or red fluorescent powders with a mass ratio of 1:9:0.05. The lattice microstructure is actuated by rotating sphere magnet at a frequency of 2 Hz. The distance between the elastomer structure on the upper surface of the sphere magnet is about 4 mm.

## Collection of sprayed aerosol
The aerosol is prepared by the ethanol solution doped with fluorescein isothiocyanate with a mass ratio of 0.13%. The microdroplets were generated by a sprayer. The viscous liquid environment is formed by silicone oil. The sphere magnet was placed beneath the strip structures and actuated at a frequency of 5 Hz to collect these microdroplets. The brightness of the fluorescence signal was calculated as $Bright = 0.299*R + 0.587*G + 0.114*B$, where $R$, $G$, and $B$ refer to the RGB value of each pixel.

## Fabrication and actuation of untethered swimming robots
A polylactic acid (PLA) flat plate fabricated by fused filament fabrication was used as substrate. To ensure a stable connection between the substrate and magneto-elastomer, the flat plate was coated by a thin layer of Ecoflex elastomer. An acrylic sheet with groove pattern was fabricated by a laser processing platform and placed on top of the flat plate. The magnetic slurry was poured onto the surface of the acrylic mold, and the entire structures were placed in a vacuum chamber for 5 min so that the elastomer could thoroughly fill the cavity portion of the acrylic mold. After degassing, the magneto-elastomer was cured in an oven (60 °C) for 20 min. The elastomer structure was peeled off from the acrylic mold and stayed on the PLA substrate. The whole structure was placed in toluene to induce the buckling transformation and are magnetized via the magnetizer. Two magnetized elastomer structures are connected together as the robotic structure via uncured silicone elastomer as the bonding agent. The untethered swimming robot was actuated by a three-axis Helmholtz coils, which generate an oscillating magnetic field with the magnetic field strength and frequency of 9 mT and 12 Hz, respectively, and were controlled by a custom-programmed software with LABVIEW.

## Fluidic mixing device
A PDMS device with Y-shape channel was fabricated. The PDMS device and the glass substrate with the elastomer strips are treated by plasma for 10 min. The PDMS device is placed on top of the glass substrate by pressing for 2 h to form stable connection. Glycerol colored with two different dyes (mass ratio: 0.01%) was pumped into the fluidic channel via peristaltic pump and formed a stratified fluid. The magnetic strips were actuated by the sphere magnet beneath the device at a frequency of 5 Hz to form turbulent flow under low Reynolds number environment.

## Simulation methods
(i) The simulations of magnetic fields generated by cuboid permanent magnet and spherical magnet were performed by COMSOL. (ii) The simulation of the buckling transformation of magneto-elastomers in solvent was performed by ABAQUS. The simulations include the parameters: the elastic modulus of the elastomer (67 kPa) and the swelling ratio. The non-freestanding structures in the simulation are assumed to be thermally expanded. The buckling transformations of the structures are simulated by setting the change of environment temperature and the coefficient of thermal expansion so that they undergo the same volume expansion as the swelling process in solvent. (iii) The simulation of the transformation under magnetic stimuli was conducted by a user-element subroutine (UEL) in ABAQUS. Considering the asymmetry of Cauchy stress induced by magnetic body torques, a customized 8-noded brick elements are employed to handle the nine components of the Cauchy stress tensor. (iv) The simulation

of the coupling stimulation of magnetic field and solvent was performed by COMSOL. Elastomer structures were divided into smaller subsections with varying magnetic torques in COMSOL structural mechanics module. The torques of subsections were recalculated according to the deformation of the elastomer structure until it reaches the equilibrium state. Subsequently, solvent stimulation was applied. The deformed elastomer structure could undergo the swelling process and buckling transformation, which are simulated through thermal expansion function.

## Characterization techniques

The magnetization profile was measured by a magneto-optical sensor (MagViewS, Matesy, Jena, Germany). The mechanical property of the magneto-elastomers was measured by a mechanical tester (MACH-1 mechanical tester v500cst, MA008). The magnetic hysteresis of the magnetic elastomer was measured by a PPMS model 6000 Quantum Design VSM.

## Reporting summary

Further information on research design is available in the Nature Portfolio Reporting Summary linked to this article.

## Data availability

All data generated in this study are provided in the article and its Supplementary Information.

## Code availability

All the relevant code in this paper is available upon request from the corresponding author.

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

## Acknowledgements

This work was supported by the Hong Kong Research Grants Council (RGC) with project nos. RFS2122-4S03, R4015-21, JLFS/E-402/18, C1134-20GF, and E-CUHK401/20; the ITF project with Project No. MRP/036/18X funded by the HKSAR Innovation and Technology Commission (ITC), the Croucher Foundation Grant with ref. no. CAS20403, and the CUHK internal grants. The authors also thank the support from the SIAT-CUHK Joint Laboratory of Robotics and Intelligent Systems, and the Multi-scale Medical Robotics Centre (MRC), InnoHK, at the Hong Kong Science Park.

## Author contributions

N.X. and D.J. conceptualized the research. N.X., D.J., C.P., and L.S. fabricated the soft structures. N.X., D.J., C.P., and J. Zhang performed the magnetic actuation experiments and analyzed the experimental data. Z.Y. conducted the simulation for magnetic flux distribution. N.X., L.W., and J. Zhao performed the simulation and theoretical studies for the active deformation under stimuli from solvent and magnetic field. L.Z., D.J. directed the project. N.X., D.J., and L.Z. prepared the manuscript. All authors discussed the results and reviewed the manuscript.

## Competing interests

The authors declare no competing interests.

## Additional information

[1]Department of Mechanical and Automation Engineering, The Chinese University of Hong Kong, Hong Kong, China. [2]Department of Biomedical Engineering, City University of Hong Kong, Kowloon, Hong Kong SAR, China. [3]CAS Key Laboratory of Mechanical Behavior and Design of Materials, Department of Modern Mechanics, University of Science and Technology of China, 230026 Hefei, Anhui, China. [4]Chow Yuk Ho Technology Center for Innovative Medicine, The Chinese University of Hong Kong, Hong Kong, China. [5]CUHK T Stone Robotics Institute, The Chinese University of Hong Kong, Hong Kong, China. [6]Department of Surgery, The Chinese University of Hong Kong, 999077 Hong Kong SAR, China. [7]Present address: School of Materials Science and Engineering, Harbin Institute of Technology (Shenzhen), Guangdong, China. ✉e-mail: jindongdong@link.cuhk.edu.hk; lizhang@mae.cuhk.edu.hk

