## [Peer Review File · Nature Communications]

Dynamic morphological transformations in soft architected materials via buckling instability encoded heterogeneous magnetizationREVIEWER COMMENTS

Reviewer #1 (Remarks to the Author):

The manuscript entitled "Dynamic topological transformations in soft architected materials via Two buckling instability encoded heterogeneous magnetization" reported a magnetic control approach for achieving dynamic topological alteration of buckling topologies. Magnetic field inputs with programmable intensities, directions, and gradients allowed bidirectional anisotropic topological transformations to be realized far beyond their semi-static states. 3D buckling structures demonstrated various switchable fluid properties due to variable structural reconfiguration based on a time-varying magnetic field. Using the dynamic modulation technique, they also demonstrated possible uses of 3D morphable architectures in fluidic actuation, object entrapment, and sensitivity-enhanced biomedical investigation. The work is interesting from the engineering perspective viewpoints. However, the novelty and the broad impacts of the work may not reach the level pursued by Nature Communications. In addition, some shortcomings in the manuscript need to be settled or improved.

1. Control and deploying the dynamic topological transformations via mechanical instability in magnetic soft composites has been widely studied and reported. The actuation mechanism via delicate magnetic field has been established. The theoretic modelling that predicts the shape transformations has been also constructed. There is no novelty in the materials and structure fabrication methods, which limits the broader impacts. The only impressiveness of the work demonstrates three applications as shown in Fig. 4 and Fig. 5. Thus, the work is more suitable for a specific journal.
2. From theoretic perspective, a study on the shape transformation under the coupling of the magnetic and solvent stimulation is interesting and innovative, while it is lacking in the current manuscript.
3. As mentioned in Lines 133 and 134, the side edges of the stripe away from the geometrical constraint will get swelled. Thus, it would be more accurate to show a slight inclination in both ends of the swelled strip.
4. In the caption of Fig. 2, it is mentioned that the structures are under the stimuli of both solvent and a magnetic field, but no trace of solvent stimulation is seen in the figure.
5. It would be recommended to add the effect of magnetic field density and direction angle on the wavelength to Fig. 2f.
6. Does the material show spatial heterogeneous response to the solvent? If yes, how this spatial heterogeneity programmed?

Reviewer #2 (Remarks to the Author):

The authors reported a method to encode continuously varying magnetization profiles in cellular microstructures and demonstrate their magnetically-induced dynamic properties and functions. The method involves swelling of microstructures made of silicone elastomer doped with NdFeB particles with organic solvents. The swelling induced buckling of the cell walls, which form a wavy profile. Subsequent magnetization of NdFeB microparticles under a uniform high magnetic field encodes a

continuously varying magnetization profile along the cell walls and enables various dynamic behaviours and functions of the cellular microstructures. The idea to use buckling generated wave patterns to program magnetization profiles is very interesting. The authors' detailed characterizations of the parameters involved in inducing buckling and forming various dynamic behaviours and functions are comprehensive. However, there are several significant issues that need to be addressed before it can be recommended for publication in Nature Comm: the conceptual contextualization of the work, the lack of insights from theoretical analysis and modelling, and the potential of applications.

1. The authors used the phrase "topological transformation" to describe the shape change of their cellular microstructures. Ref 1 (Nature 592, 386-391 (2021)) seems to be the main reference for this term. The topology of a structure, mathematically speaking, denotes the connectivity of the components of the structure. In Ref. 1, the connectivity of the cellular microstructures was changed, for example, from six-neighbour connections (hexagonal lattice) to three-neighbour connections (triangular lattice). Hence it is suitable for them to call their structural change "topological transformation". However, in the present manuscript, the buckling of cell walls does not change the connectivity of the microstructures, so it seems incorrect to call their structure change "topological transformation".

2. The authors performed comprehensive characterizations of the parameters involved in buckling (Fig. 2). These studies appear to be phenomenological studies only. In Fig. 3, the authors have used FEM method to compare experimental observations with simulations. These comparisons provide some mechanistic understanding. Why not do similar modelling for the results in Fig. 2? Moreover, aside from reproducing experimental patterns, are there any insights or understandings that could be obtained from these modelling studies? It will be good to state them explicitly.

3. The two functions demonstrated, the transport and merging of droplets and the selective trapping, directly involve the interaction between objects being manipulated and underlying microstructures. However, the fluorescent signal enhancement does not appear to involve direct interaction between the aerosol droplets and the microstructure, so one would ask, why do we need this particular type of dynamic transformation to cause the disturbance that enables the aerosol droplet merging? The authors need to think about the unique properties and functions of the dynamic transformations of their microstructure and explore how they can be used.

There are a few more minor issues:

1. The methods section needs to include more detailed experimental procedures for each of their experiments shown in the figures and supplementary figures. At the moment, they are too general.

2. Fig. 3a, calling these different connectivity types "boundary conditions" seems to be another misnomer.

3. Line 133 – 134, the microscopic swelling of silicone elastomer, the infusion of solvent in polymer chains, should occur throughout the structure, even though the extent of the swelling may not be observable by light microscopy.

Reviewer #3 (Remarks to the Author):

The manuscript presents very interesting way to fabricate dynamically reconfigurable structures employing buckling and magnetic actuation. Specifically, non-freestanding composite structure of elastomer with magnetic particles. The structure is firstly buckled by solvent swelling. while maintaining the buckled state, external magnetic field is applied to 'imprint' the magnetic flux orientation of those embedded magnetic particles. After de-swelling, the structure returns to its initial flat shape. And this shape can be changed into various shapes by applying external magnetic field, thanks to the imprinted magnetic flux in magnetic particles. Using this principle, the authors have demonstrated the generation of fluid flow by dynamic modulation of the shape with external magnetic field. And, selective particle trapping and sensitive droplet detection have also been shown using the suggested structures.

Overall, the manuscript seems quite interesting, but following questions/suggestions should be considered while preparing their revised version of the manuscript.

First of all, I don't find any explanation on the mechanism and theoretical analysis on the observed phenomenon. The authors have shown FEM results only, without discussing involved mechanics and magnetomechanical coupling.

Second, I wonder the buckled shape be critically dependent on the boundary conditions (Fig. 3). Is this true? This dependence is reproducible, i.e., the observed buckling shape was always observed under fixed experimental condition?? Boundary condition I, for example, the central part buckles into a complete wave (1 full wavelength), then the full wave can be either down-up or up-down, I think. This is because both the buckled shape seems to be in the same energy state, in terms of energy approach. This is true for other cellular structures.

Regarding the selective particle trap, the experimental set-up seems not be correct: for opening of the cellular structure of 500um, small (200um) particle and large (600um) particle mixture seems irrational. The authors should select the particle sizes just above/below the deformed size of cellular structure. For fluidic flow generation experiments, the authors should give explicit power consumed to generate the flow, and discuss the result to other flow generation methods in terms of energy cost. Further, rather than those simple flow generation, they should show unique capability of their system, compared to typical methods. For example, I think the present flow generation may be much informative when it can be used to fluid mixing in microfluidic channel flow: this micro-scale fluid mixing is very costly, while one can generate kind of secondary flow using method proposed in this work to mix different fluids efficiently.

Overall, I think the manuscript can be published once the authors revise their manuscript properly.

Response to Reviewer #1

The manuscript entitled "Dynamic topological transformations in soft architected materials via two buckling instability encoded heterogeneous magnetization" reported a magnetic control approach for achieving dynamic topological alteration of buckling topologies. Magnetic field inputs with programmable intensities, directions, and gradients allowed bidirectional anisotropic topological transformations to be realized far beyond their semi-static states. 3D buckling structures demonstrated various switchable fluid properties due to variable structural reconfiguration based on a time-varying magnetic field. Using the dynamic modulation technique, they also demonstrated possible uses of 3D morphable architectures in fluidic actuation, object entrapment, and sensitivity-enhanced biomedical investigation. The work is interesting from the engineering perspective viewpoints. However, the novelty and the broad impacts of the work may not reach the level pursued by Nature Communications. In addition, some shortcomings in the manuscript need to be settled or improved.

Response: We sincerely thank the reviewer for the valuable comments. Based on your questions, we have added corresponding experiments, discussions and changes in the revised manuscript, which we hope can address your concerns on the novelty and impact of our work. Please check the following point-by-point response for details.

1. Control and deploying the dynamic topological transformations via mechanical instability in magnetic soft composites has been widely studied and reported. The actuation mechanism via delicate magnetic field has been established. The theoretic modelling that predicts the shape transformations has been also constructed. There is no novelty in the materials and structure fabrication methods, which limits the broader impacts. The only impressiveness of the work demonstrates three applications as shown in Fig. 4 and Fig. 5. Thus, the work is more suitable for a specific journal.

Response: Thanks for your comment. There are some reports that utilize buckling instability to promote the deformation of magneto-elastomers, such as the control of the deformation of bistable conical shells using magnetic forces or the adjustment of the buckling strength of uniformly magnetized spherical shells with external magnetic field [1. Chen, Tian, Mark Pauly, and Pedro M. Reis. "A reprogrammable mechanical metamaterial with stable memory." *Nature* 589.7842 (2021): 386-390; 2. Yan, Dong, et al. "Magneto-active elastic shells with tunable buckling strength." *Nature Communications* 12.1 (2021): 2831.]. Ref. 1 reports a conical shell composed of magneto-elastomer which can be independently and reversibly switched between bistable states by electromagnetic coil actuation, and is used as a physical binary unit. As each physical unit exhibits distinct mechanical properties under an ON-OFF magnetic field, an array structure constructed from these physical units exhibits widely tunable stiffness and strength. Ref. 2 reports a mechanism for tuning the buckling strength of thin

spherical shells using the coupling of mechanics and magnetism. For spherical shells made of hard magnetic materials, the ratio between the maximum buckling pressure and critical buckling pressure is regulated by adjusting the magnitude and direction of the external magnetic field. The general mechanism of coupling elastic deformation and magnetic actuation is expected to open an axis in the design of mechanical systems with tunable mechanical properties. Previous studies focused on mechanical or metamaterial systems constructed by magneto-elastomers with relatively uniform magnetization profiles, which may limit the degree of deformation freedom and the corresponding functionalities.

In this study, we aim to present dynamic morphological transformation in soft architected materials with 3D heterogeneous magnetization profiles encoded by buckling instability. To unlock active shape deformation of morphable structures, control strategies based on chemical solvent, temperature, pH, and light have been developed before. However, the reported regulation methods are mostly based on the quasi-static modulation of architected materials, and it is still challenging to develop a highly controllable strategy enabling the dynamic regulation and multimodal anisotropic transformations crossing different scales. The magnetic regulation strategy open possibilities to address the limitations due to its excellent controllability and programmability. To explore complex transformation of magnetoactive materials, elaborate magnetization programming techniques are often required. In this work, we propose a facile magnetization programming method by using the buckling instability phenomenon, which provides a template-free and on-demand manner for the formation of magnetoactive materials with 3D magnetization profile. Through programming magnetic field inputs including magnetic field direction, strength, and gradient, the buckling-induced magnetized elastomers could exhibit multi-mode anisotropic transformations or robust switch between different buckling states, thereby providing a flexible strategy to dynamically regulate the morphologies in soft structures and exhibiting promising potentials in the field of fluidic manipulation, selective particle trapping, soft robotics, and sensitivity-enhanced biomedical analysis.

Based on your comment, we highlight the novelty and impact of this work in the revised manuscript. In addition, we have supplied the following studies: (1) the coupling stimulation of solvent and magnetic field on the elastomer structure; (2) modelling studies on the solvent and magnetic responsive behaviors of the magnetoactive material; (3) Further application demonstrations for microfluidic device and wireless soft robot. We hope such revision can address your concerns at a satisfactory level and convince you of the suitability of our work for *Nature Communications*.

2. From theoretic perspective, a study on the shape transformation under the coupling of the magnetic and solvent stimulation is interesting and innovative, while it is lacking in the current manuscript.

Response: Thank you for the question. We have supplemented the study on the effects of coupled magnetic and solvent stimulation on structural deformation in the revised work. Various factors including magnetization, solvent environments, and stimulation sequences, combined with modelling studies are analyzed in detail. According to your comment, we have added the following discussion in the “**2.3 Geometric transformation of magneto-elastomer under the coupling stimulation**” part and “**2.4 Morphological transformations of cellular structures**” part of the revised manuscript.

(1) The impact of the coupled stimulation on the strip structures with different magnetization types and solvent environments is first studied. Here, the magnetization profiles of magneto-elastomer are divided into two types, i.e., buckling instability-encoded magnetization (BM) and uniform magnetization (UM), while the solvent environment includes deionized (DI) water and organic solvent toluene. So, there are 4 coupled stimulus types as shown in Figure R1.1(a).

For magneto-elastomers that adopt buckling encoded magnetization, waveforms will be formed following the change of the external magnetic field direction whatever the solvent environment is. Under the stimulation of organic solvent, the structure is always in a swelling-induced buckling state. To quantitatively describe the morphological changes of the structures upon the coupled stimulation, a phase shift parameter $\phi = \frac{\Delta x}{\lambda} \times 360$ is defined, where Δx and λ represent the displacement of the position of the peak and the wavelength, respectively, as shown in Figure R1.1(c). For BM structures, the phase shift varies with the direction angle of the magnetic field, resulting in a geometrical transformation in the form of traveling waves. Alternating the solvent stimulation to DI water only decreases the deformation degree, yet does not influence the deformation form.

In contrast, for the uniformly magnetized structure, relatively few deformation modes can be generated under the coupled stimulation of magnetic field and toluene. However, when immersed in DI water, the structure only exhibits bending deformation. By adjusting the direction of magnetic stimulation, the bending peak position will move back and forth within a limited distance along the length direction, leading to the change of deformation form, as shown in Supplementary Movie 2.

We develop finite element analysis (FEA) model to verify the buckling modes of the structures under the coupled stimulation and their phase shift with varied magnetic field direction, as shown in Figure R1.1(c). The elastomeric structures in the FEA model are assumed to be thermally expanded in response to environment temperature. Simultaneously, external magnetic torque loads ($\mathbf{T} = \mathbf{RM} \times \mathbf{B}$, where \mathbf{M} and \mathbf{B} represent the magnetization profile and external magnetic field, respectively. \mathbf{R} is a rotational matrix determined by the structure’s rotational deflection) are applied to simulate the magnetic stimulus, and the magnetic loads are calculated and updated in real time according to the deformation of the structures. The modelling results demonstrate the impact of different magnetic field directions on the geometrical transformation under the coupled stimulation.

The results of the coupled stimulation illustrate that the magnetic control strategy enables us to explore controllable buckling deformation of 3D morphable structures, and the buckling instability encoded magnetization endows the soft structures with more degrees of deformation freedom, which provides great potentials in the areas like droplet manipulation, microfluidic devices and soft robotics.

Figure R1.1. Geometric transformation of magneto-elastomer under the coupling stimulation of magnetic field and solvent. **a** Impact of different magnetization profiles and solvents on the deformation of strip structure. BM: buckling instability encoded magnetization. UM: Uniform magnetization. Magnetic field strength: 150 mT. **b** Variation of phase shift with different magnetic field directions and stimuli types. **c** Simulation results for strip with buckling encoded magnetization profile under the coupling stimulation of magnetic field and organic solvent. The scale bars are 3 mm.

(2) The regulation of buckling states via magnetic stimulation and the impact of stimulation sequences on morphological transformation are investigated. In this part, magneto-elastomer with buckling instability-encoded magnetization is used. As shown in Figure R1.2(a), compared to the buckling state I induced by toluene alone, the application of coupled stimulation triggers the change to buckling state II with different deformation morphology.

Such geometric transformation can be maintained even when the magnetic field is removed. The result indicates that external magnetic stimulation can help to determine the stable state of soft structure during buckling process, and moreover, this regulation does not require prolonged magnetic field stimulation input.

Secondly, the impact of stimulation sequences is studied as shown in Figure R1.2(b). When a weak magnetic field (20 mT) is applied, different stimulation sequences would affect the deformation results. If the magnetic stimulation is applied first followed by the solvent stimulation, the deformation shapes of the elastomeric structure can be dynamically changed by tuning the magnetic field direction. If the solvent stimulation is applied first followed by magnetic field stimulation, only one deformation shape can be obtained no matter what the field direction. In comparison, when the magnetic field is as strong as 150 mT, it can be found that the stimulation sequences have a negligible effect on the deformation results. At this time, the buckling morphologies of soft structures are always affected by the magnetic field direction.

The impacts of magnetic field strength and stimulation sequences can be explained by the stored strain energy in the elastomeric structures. For the case of applying magnetic stimulation first, the stored strain energy of the structure in the initial state can be ignored, and the weak magnetic field is sufficient to affect the morphology of the structure. The magnitude of structural deformation after solvent stimulation increases gradually during the swelling process. For the case of applying solvent stimulation first, since the structure is in a swollen and buckling state, the stored strain energy would be relatively large, so that the input of weak magnetic field cannot provide sufficient energy to overcome the energy barrier for geometric transformation.

Figure R1.2. Geometric transformation of strip structures under different sequences of solvent and magnetic stimulation. **a** Different buckling states of the strip induced by the coupled stimulation. **b** Deformation of strip structures under different stimulation sequences and magnetic field strength. The scale bars are 3mm.

The impact of the coupled stimulation on the deformation of lattice structure is also studied. By applying magnetic stimulation first followed by the solvent stimulation, the structure can form different buckling states with the change of the direction of the external magnetic field, as shown in Figure R1.3(a). While for the case of applying solvent stimulation first, the external magnetic stimulation cannot provide sufficient energy to change its morphology. In addition, as shown in Figure R1.3(b) and (c), the stable state of lattice structure during buckling process can also be regulated via the magnetic stimulation, all of which are consist with the results of aforementioned strip structure.

Figure R1.3. Geometric transformation of lattice structures under different sequences of solvent and magnetic stimulation. **a** Deformation of lattice structures under different stimulation sequences. **b-c** Different buckled states of the lattice structures induced by the coupling stimulation of magnetic fields and solvent. The scale bars are 5 mm.

Generally, the buckling modes exhibited by soft structures are multi-stable [1. Bertoldi, Katia, et al. "Flexible mechanical metamaterials." *Nature Reviews Materials* 2.11 (2017): 17066; 2. Pal, Aniket, et al. "Exploiting mechanical instabilities in soft robotics: control, sensing, and actuation." *Advanced Materials* 33.19 (2021): 2006939.] and the final state after buckling process possesses a certain randomness. The study on stimulation sequence in our work demonstrates that the introduction of magnetic stimulation could help to predetermine the buckling states of soft architected materials, and enable the formation of definite and controllable buckling states without prolonged magnetic stimulation input. Therefore, the regulation strategy provides a promising way for the study on mechanically unstable systems and the modulation of chirality and patterning of the cellular structures.

3. As mentioned in Lines 133 and 134, the side edges of the stripe away from the geometrical constraint will get swelled. Thus, it would be more accurate to show a slight inclination in both ends of the swelled strip.

Response: Thank you for the comment. We have revised the schematic image of swelled strip according to your suggestion and experimental result, which shows slight inclination in both ends of the swelled strip. Revised image is also provided here as Figure R1.4 for your reference.

Figure R1.4. Schematics of dynamic geometrical transformation of magneto-elastomers.

4. In the caption of Fig. 2, it is mentioned that the structures are under the stimuli of both solvent and a magnetic field, but no trace of solvent stimulation is seen in the figure.

Response: Thank you for the question. We are sorry for the confusion caused. In the initial manuscript, the structures are under the sequential applications of solvent and magnetic stimulation, and these two stimuli are not simultaneous. In the revised manuscript, the coupling stimulation of magnetic field and solvent as you suggested is studied. To avoid misunderstanding, the caption of new Fig. 2 is carefully organized as shown in Figure R1.5.

Figure R1.5. Buckling instability encoded heterogeneous magnetization in strip magneto-elastomer. **a** Demonstration of the original strip structure, its buckled state, and geometric transformation. The magnetic flux density profiles of the structure before and after magnetization are measured by magneto-optical sensor. The modeling results show the deformation of the strip structure under the stimuli from solvent and external magnetic field, and the magnetization profiles of the strip. The scale bar is 5 mm. **b-d** The quantitative relationship between wavelength, amplitude of the strip structure and the geometric parameters of the structure including length (**b**), height (**c**), thickness (**d**). The gray area in Fig. 2b represents no wavy pattern generated in the elastomeric structure. **e-f** The quantitative relationship between the magnetic field strength (**e**), direction angle of magnetic field and amplitude of the buckling structure (**f**). The inset images illustrate experimental and simulation results. The inset image with label '1' shows twisted strip structures when the direction angle is equal to 90° . The inset image with label '2' shows collapsed strip structures with the height of 5.0 mm when the direction angle is equal to 180° . The inset image with label '3' shows the wavy patterns of strip structures with the thicknesses of 1.6 mm when the

direction angle is equal to 180° . Error bars stand for the standard error of the mean with the number of trials $n = 3$.

5. It would be recommended to add the effect of magnetic field density and direction angle on the wavelength to Fig. 2f.

Response: Thank you for the suggestion. We have evaluated the effect of magnetic field strength and direction angle on the wavelength of strip structure, as shown in Figure R1.6. Buckling instability-encoded magnetized strip structures with different heights are fabricated for test. In Figure R1.6(a), the wavelength of magneto-elastomer almost keeps constant under different magnetic field strength, indicating the magnitude of magnetic torque only affects the deformation amplitude rather than wavelength. On the other hand, in Figure R1.6(b), when the direction angle of magnetic field is between 90° and 270° , collapse or torsion of the magnetic strips may occur, especially for the strip structure with larger height, leading to the disappearance of wavy pattern. But once the wave forms in magneto-elastomers, it is found that the adjustment of magnetic field direction angle cannot induce any obvious change in the wavelength. Therefore, we conclude that both magnetic field density and direction angle have no effect on the wavelength, and these results are added as Supplementary Fig. 6 in the revised supporting information.

Figure R1.6. Impact of magnetic field strength and direction angle on the wavelength. **a** Effect of magnetic field strength on the wavelength. **b** Effect of direction angle on the wavelength.

6. Does the material show spatial heterogeneous response to the solvent? If yes, how this spatial heterogeneity programmed?

Response: Thank you for the question. The magneto-elastomers used in this work do not exhibit heterogeneous response to the solvent, as shown in Figure R1.7. Heterogeneous deformation in the elastomer structure along height direction is achieved because the bottom of elastomer is constrained during the swelling process, while the other end is in free state. We can adjust or control the morphological transformation of elastomers by introducing defects in appropriate positions of the elastomers during manufacturing process. As shown in Figure R1.8, by arranging the conical defects on the surface of the elastomer, different waveforms can be achieved. According to your comment, we have added Figure R1.8 in the revised supporting information as Supplementary Fig. 15.

Figure R1.7. Swelling and deswelling processes of magneto-elastomer in toluene and ethanol, indicating the spatial homogeneous response of free material to the solvent.

Figure R1.8. Geometric transformation of strip structures with cut patterns under solvent stimulation. **a** Schematic diagram of the cut patterns and transformation process. **b** Experimental results of the transformation of strip structure with cut patterns. The scale bar is 2 mm.

Response to Reviewer #2

The authors reported a method to encode continuously varying magnetization profiles in cellular microstructures and demonstrate their magnetically-induced dynamic properties and functions. The method involves swelling of microstructures made of silicone elastomer doped with NdFeB particles with organic solvents. The swelling induced buckling of the cell walls, which form a wavy profile. Subsequent magnetization of NdFeB microparticles under a uniform high magnetic field encodes a continuously varying magnetization profile along the cell walls and enables various dynamic behaviours and functions of the cellular microstructures. The idea to use buckling generated wave patterns to program magnetization profiles is very interesting. The authors' detailed characterizations of the parameters involved in inducing buckling and forming various dynamic behaviours and functions are comprehensive. However, there are several significant issues that need to be addressed before it can be recommended for publication in Nature Comm: the conceptual contextualization of the work, the lack of insights from theoretical analysis and modelling, and the potential of applications.

Response: Thank you very much for your feedback. We have answered every comment and made corresponding changes in the revised manuscript. Please check the following point-by-point response for details.

Comment 1. The authors used the phrase “topological transformation” to describe the shape change of their cellular microstructures. Ref 1 (Nature 592, 386-391 (2021)) seems to be the main reference for this term. The topology of a structure, mathematically speaking, denotes the connectivity of the components of the structure. In Ref. 1, the connectivity of the cellular microstructures was changed, for example, from six-neighbour connections (hexagonal lattice) to three-neighbour connections (triangular lattice). Hence it is suitable for them to call their structural change “topological transformation”. However, in the present manuscript, the buckling of cell walls does not change the connectivity of the microstructures, so it seems incorrect to call their structure change “topological transformation”

Response: Thank you for sharing your comments. In the previous manuscript, the “topological transformation” is used to refer to the change of geometrical shape and surface morphology rather than its mathematical meaning. To avoid misunderstanding, we have changed the term “topological transformation” to “morphological transformation” in the revised manuscript according to your suggestion.

Comment 2. The authors performed comprehensive characterizations of the parameters involved in buckling (Fig. 2). These studies appear to be phenomenological studies only. In Fig. 3, the authors have used FEM method to compare experimental observations with simulations. These comparisons provide some mechanistic understanding. Why not do similar modelling for the results in Fig. 2? Moreover, aside from reproducing experimental patterns, are there any insights or understandings that could be obtained from these modelling studies? It will be good to state them explicitly.

Response: Thanks a lot for your comment. According to your suggestion, we have supplemented the simulation results for strip magneto-elastomers in Figure 2 in the revised manuscript, which are also provided here as Figure R2.1.

Figure R2.1. Buckling instability encoded heterogeneous magnetization in strip magneto-elastomer. **a** Demonstration of the original strip structure, its buckled state, and geometric transformation. The magnetic flux density profiles of the structure before and after magnetization are measured by magneto-optical sensor. The modeling results show the deformation of the strip structure under the stimuli from solvent and external magnetic field, and the magnetization profiles of the strip. The scale bar is 5 mm. **b-d** The quantitative relationship between wavelength, amplitude of the strip structure and the geometric parameters of the structure including length (**b**), height (**c**), thickness (**d**). The gray area in Fig. 2b represents no wavy pattern generated in the elastomeric structure. **e-f** The quantitative relationship between the magnetic field strength (**e**), direction angle of magnetic field and amplitude of the buckling structure (**f**). The inset images illustrate experimental and

simulation results. The inset image with label ‘1’ shows twisted strip structures when the direction angle is equal to 90° . The inset image with label ‘2’ shows collapsed strip structures with the height of 5.0 mm when the direction angle is equal to 180° . The inset image with label ‘3’ shows the wavy patterns of strip structures with the thicknesses of 1.6 mm when the direction angle is equal to 180° . Error bars stand for the standard error of the mean with the number of trials $n = 3$.

In our work, finite element modelling studies could help to clarify the influences of various factors (e.g., geometric dimension, defect distribution, connectivity type, etc.) on the morphological transformations, which is a potential tool to further guide the structural design and predict deformation pattern of magneto-elastomers. For example, the impact of connectivity types with different degrees of freedom on structural deformation is explored with the help of FEA model as shown in Discussion R1. Besides, to broaden the role of the modelling studies and elucidate the relevant mechanical mechanisms, we supplement analytical models in Discussion R2, which could give the quantitative relationship between the key parameters of structural deformation and geometric parameters, and provide a theoretical guideline for the structural design. Furthermore, the developed model could help to explore the possible deformation modes of buckling structures in regard of the random buckling process and then guide the experimental results.

Discussion R1: The modelling studies provide effective tools to analyze the effects of nodal degrees of freedom on structural deformation. In section 2.3 “Transformations of cellular structures” and Figure 3, we analyze the impact of different connectivity types on structural transformation, where these connections would restrict the translation at both ends of the strip, but do not affect its rotation. In addition, we could study structures with different degrees of freedom via the modelling method. As shown in Figure R2.2, the two sides of the plate structure are distributed with non-responsive material and fixed constraints, which restricts the rotation at the ends. With such restrictions, the structure shows a bell-shaped deformation under solvent stimulation which is different from that with free-to-rotate junctions. Asymmetric deformation would be formed by adopting mix connectivity types. Therefore, the modelling methods provide detailed analysis of the impact of different constraint types, which is conducive to the development of more complex geometric transformations. The realization of these structures with complex constraint types can be achieved with the help of advanced fabrication methods such as multi-material 3D printing technology [*Mueller, Jochen, Jennifer A. Lewis, and Katia Bertoldi. "Architected Multimaterial Lattices with Thermally Programmable Mechanical Response." *Advanced Functional Materials* 32.1 (2022): 2105128.*]. According to your comment, we have added Figure R2.2 in the revised manuscript as Supplementary Fig. 10.

Figure R2.2. Impact of different connectivity types on the geometric transformation. **a** Strip structure with free-to-rotate junctions. **b** Strip structure with fixed junctions. **c** Strip structure with both fixed and free-to-rotate junctions.

Discussion R2: In this part, beyond FEA modeling, we supplement the analytical modeling of this work including (1) swelling-induced buckling behaviors, (2) buckling encoded 3D heterogeneous magnetization profile, and (3) magnetic response behavior. According to your comment, we have added the following discussion in the in the “**2.2 Characterization of magnetic and solvent responsive behavior of surface-attached structures**” part and “**2.4 Morphological transformations of cellular structures**” part of the revised manuscript and “**Supplementary Note 1 Analytical model**” part of the revised supporting information.

R2.1: Swelling-induced buckling behaviors

Figure R2.3. Schematic diagram of buckling transformation. **a** Transformation of elastomer strip. **b** Transformation of elastic plate with free-to-rotate junctions.

A thin elastomer strip has length L , height H , and thickness T as shown in Figure R2.3 (a). The bottom ($z=0$) and top ($z=H$) edge of the elastomer strip are fixed and free, respectively. Foppl-von Karman equations were used to describe the deformation of thin flat plates [1. Jiang, Ruiqi, Jianliang Xiao, and Jizhou Song. "Buckling of thin gel strip under swelling." *Theoretical and Applied Mechanics Letters* 7.3 (2017): 134-137; 2. Kang, Sung Hoon, et al. "Buckling-induced reversible symmetry breaking and amplification of chirality using supported cellular structures." *Advanced Materials* 25.24 (2013): 3380-3385.], as shown in Eqs. (R2.1) and (R2.2).

$$\delta(y, z) = \frac{A}{2} \times f(z) \times \sin\left(\frac{2\pi}{\lambda} y\right) \quad (\text{R2.1})$$

$$\frac{\partial^4 \delta}{\partial y^4} + 2 \frac{\partial^4 \delta}{\partial y^2 \partial z^2} + \frac{\partial^4 \delta}{\partial z^4} + \frac{t_{11}}{D} \frac{\partial^2 \delta}{\partial y^2} = 0 \quad (\text{R2.2})$$

$\delta(y, z)$ represents the out-of-plane displacement of the strip. $u_y(y, z)$ and $u_z(y, z)$ are the in-plane displacement along the y and z direction, respectively. λ represents the wavelength. t_{11} and D are the membrane force along the y direction and bending stiffness, respectively. A is the peak-to-peak value. $f(z)$ represents a normalized function, where $f(H) = 1$. By solving the ordinary differential equation for $f(z)$, a general solution can be obtained as Eq. (R2.3).

$$f(z) = C_1 e^{-\alpha z} + C_2 e^{\alpha z} + C_3 \cos(\beta z) + C_4 \sin(\beta z) \quad (\text{R2.3})$$

$$\alpha = \sqrt{\sqrt{\frac{t_{11}}{D} \left(\frac{2\pi}{\lambda}\right)^2} + \left(\frac{2\pi}{\lambda}\right)^2} \quad (\text{R2.4})$$

$$\beta = \sqrt{\sqrt{\frac{t_{11}}{D} \left(\frac{2\pi}{\lambda}\right)^2} - \left(\frac{2\pi}{\lambda}\right)^2} \quad (\text{R2.5})$$

The values of C_1, C_2, C_3, C_4 can be determined by the boundary conditions shown in Figure R2.3. For the strip structure in Figure R2.3(a), the clamped edge and the free edge of the strip determine the equations:

$$\left\{ \begin{array}{l} f(0) = 0 \\ \frac{\partial f}{\partial z} \Big|_{(z=0)} = 0 \\ \left(\frac{\partial^2 \delta}{\partial z^2} + \frac{\nu \partial^2 \delta}{\partial y^2} \right) \Big|_{(z=H)} = 0 \\ \left[\frac{\partial^3 \delta}{\partial z^3} + (2 - \nu) \frac{\partial^3 \delta}{\partial y^2 \partial z} \right] \Big|_{(z=H)} = 0 \end{array} \right. \quad (\text{R2.6})$$

, where ν is the Poisson's ratio. For the strip structure in Figure R2.3(b), the free-to-rotate junctions determine the equations.

$$\left\{ \begin{array}{l} \delta(0, z) = \delta(L, z) = 0 \\ \left(\frac{\partial^2 \delta}{\partial y^2} + \nu \frac{\partial^2 \delta}{\partial z^2} \right) \Big|_{(y=0)} = \left(\frac{\partial^2 \delta}{\partial y^2} + \nu \frac{\partial^2 \delta}{\partial z^2} \right) \Big|_{(y=L)} = 0 \end{array} \right. \quad (\text{R2.7})$$

With the boundary conditions shown in Eq. R2.6, we can get the equations for C_1, C_2, C_3, C_4 .

$$-\alpha C_1 + \alpha C_2 + \beta C_4 = 0 \quad (\text{R2.8})$$

$$C_1 + C_2 + C_3 = 0 \quad (\text{R2.9})$$

$$\mathbf{A}_{co} \begin{bmatrix} C_3 \\ C_4 \end{bmatrix} = \begin{bmatrix} A_{co11} & A_{co12} \\ A_{co21} & A_{co22} \end{bmatrix} \begin{bmatrix} C_3 \\ C_4 \end{bmatrix} = \begin{bmatrix} 0 \\ 0 \end{bmatrix} \quad (\text{R2.10})$$

$$\text{Det}(\mathbf{A}_{co}) = 0 \quad (\text{R2.11})$$

where,

$$A_{co11} = \left(\beta^2 + \nu \frac{4\pi^2}{\lambda^2} \right) \cos(\beta H) + \left(\alpha^2 - \nu \frac{4\pi^2}{\lambda^2} \right) \cosh(\alpha H)$$

$$A_{co12} = \left(\beta^2 + \nu \frac{4\pi^2}{\lambda^2} \right) \sin(\beta H) + \beta \left(\alpha - \nu \frac{4\pi^2}{\alpha \lambda^2} \right) \cosh(\alpha H)$$

$$A_{co21} = \left(-\beta^3 + (2 - \nu) \beta \frac{4\pi^2}{\lambda^2} \right) \sin(\beta H) + \left(\alpha^3 - (2 - \nu) \alpha \frac{4\pi^2}{\lambda^2} \right) \sinh(\alpha H)$$

$$A_{co22} = \left(\beta^3 + (2 - \nu) \beta \frac{4\pi^2}{\lambda^2} \right) \cos(\beta H) + \left(\alpha^2 \beta - (2 - \nu) \beta \frac{4\pi^2}{\lambda^2} \right) \cosh(\alpha H)$$

As shown in Eq. R2.10, linear homogeneous equations determine the relationship between C_3 and C_4 . To obtain nonzero solution for the equations, the determinant of the coefficient matrix is equal to zero, i.e., $\text{Det}(\mathbf{A}_{co})=0$, which gives the relationship between t_{11} and λ and determines the critical swelling condition.

Firstly, we analyze the case with free-to-rotate junctions (shown in Figure R2.3(b)). The relationship between λ and L can be determined by $W_n = L/\lambda_n$, where $2W_n$ is an integer and

represents the deformation mode of strip with the junctions. By giving the value of W_n and solving Eq R2.11, we can calculate the critical force t_{11}^c , and the relationship between the normalized critical force ($\frac{H^2 t_{11}^c}{ET^3}$) and geometry parameters, as shown in Figure R2.4. In the theoretical study, we fix the values of H and T , and study the impact of aspect ratio (L/H) on the deformation mode W_n . The theoretical and experimental results show that the relationship between L/H and W_n is in a form of step function. When the L/H reaches a critical value, the mixed deformation modes would be exhibited as shown in Figure R2.4(b) ($L/H = 3.70, L/H = 7.41$). Mixed deformation mode results for other critical conditions can be verified by extensive swelling tests.

Figure R2.4. Impact of aspect ratio on the deformation mode of the strip with free-to-rotate junctions. **a** The relationship between the normalized critical force and aspect ratio under different deformation modes. Discrete dots in orange color represents the experimental results which can also be found in Fig. R2.4b. The theoretical results illustrate the deformation modes that the strip will exhibit when a critical force is reached during swelling for a given aspect ratio. The intersection of the two curves indicates that multiple deformation modes would occur with this geometry parameter. **b** Theoretical and experimental results for the relationship between the deformation mode (W_n) and the aspect ratio. The theoretical results show a relationship in the form of a step function, which indicates that when the aspect ratio is within a certain range, the deformation mode of the strip is fixed; when the aspect ratio reaches a critical value, the strip would exhibit mixed deformation modes. As the experimental results with $L/H = 3.70$, W_n can be equal to 1.0 or 1.5. When $L/H = 7.41$, W_n can be equal to 2.0 or 2.5. The scale bar is 2 mm.

Secondly, we analyze the case in Figure R2.3(a). By solving the Eq R2.11 numerically under given λ , we can calculate the value of t_{11} and gives the relationship between λ and t_{11} . We still use the parameter $W_n = L/\lambda_n$ to illustrate the deformation of the strip. While, in this case, W_n can be any positive real number. As shown in Figure R2.4(a), for a given W_n , there is a nonlinear relationship between L/H and normalized critical force. The critical deformation mode $W_{nc} = L/\lambda_{nc}$ can be calculated according to the extreme point of the curve. Therefore, we could obtain the relationship between W_{nc} and L/H , as shown in Figure R2.5, which shows a linear correlation between W_{nc} and L/H . Through a linear fitting, λ_{nc} can be obtained, as shown in Eq. R2.15, which demonstrates that the length of the strip has negligible effect on the wavelength of the buckled strip. With the value of λ and t_{11} , the results of parameters including $C_1, C_2, C_3, C_4, \alpha$, and β can be obtained.

$$\lambda_{nc} \approx 3.256H \quad (\text{R2.15})$$

The amplitude of the out-of-plane displacement (δ) can be obtained via the minimization of the total energy [Jiang, Ruiqi, Jianliang Xiao, and Jizhou Song. "Buckling of thin gel strip under swelling." *Theoretical and Applied Mechanics Letters* 7.3 (2017): 134-137.], as shown in Eqs. R2.16 and R2.17.

$$U = \frac{0.05ET^3A^2}{H^2(1-\nu^2)} - \frac{0.0434ETA^2(\chi-1)}{\chi H(1-\nu^2)} + \frac{0.3745ETH(\chi-1)^2}{\chi^2(1-\nu^2)} + \frac{0.0029ETA^4}{H^3(1-\nu^2)} \quad (\text{R2.16})$$

$$\frac{\partial U}{\partial A} = 0 \quad (\text{R2.17})$$

U represents the total energy of the strip and χ represents the swelling ratio. The values of the peak-to-peak amplitude (A) can be calculated.

$$A = 2.724H \sqrt{\frac{\chi-1}{\chi} - 1.155 \frac{T^2}{H^2}} \quad (\text{R2.18})$$

Experimental results were employed to validate the theoretical model of the buckled strip, as shown in Figure. R2.5. The agreement between the experimental and theoretical results validates the model which could quantitatively describes the relationship between the buckling results and the geometry parameters.

Figure R2.5. Theoretical and Experimental results for the deformation of the buckled strip. **a** Theoretical results and fitting for the relationship between W_{nc} and the aspect ratio (L/H). **b** The impact of the height of the strip on the wavelength. **c** The impact of the height of the strip on the amplitude. **d** Comparison of the deformed shape of the strip structure.

R2.2: Buckling instability encoded 3D heterogeneous magnetization profile

Figure R2.6. Schematic diagram of the magnetization process and the change of magnetization profile during the recovery process.

A theoretical model was developed to describe the formation of 3D heterogeneous magnetization profile. As shown in Figure R2.6, $\mathbf{M}'(x + \delta, y + u_y, z + u_z)$ and $\mathbf{M}(x, y, z)$ represent magnetization vectors under deformation and after recovery, respectively. Considering the recovery process, we can derive the relationship between \mathbf{M}' and \mathbf{M} , as shown in Eq. R2.19.

$$\mathbf{M}(x, y, z) = \frac{\mathbf{R}}{J} \mathbf{M}'(x + \delta, y + u_y, z + u_z) \quad (\text{R2.19})$$

$J = \det(\mathbf{F})$, where \mathbf{F} is the deformation gradient of an element of body shown in Figure R1.4. \mathbf{R} is the rotational component of \mathbf{F} . To estimate the magnetization vector after recovery, we focus on the rotation of the element along z axis and y axis with Eqs. R2.20-R2.22.

$$\mathbf{R} = \begin{bmatrix} \cos(\psi) & 0 & \sin(\psi) \\ 0 & 1 & 0 \\ -\sin(\psi) & 0 & \cos(\psi) \end{bmatrix} \times \begin{bmatrix} \cos(\theta) & -\sin(\theta) & 0 \\ \sin(\theta) & \cos(\theta) & 0 \\ 0 & 0 & 1 \end{bmatrix} \quad (\text{R2.20})$$

$$\theta = \arctan\left(\frac{\partial \delta}{\partial y}\right) = \arctan\left\{\frac{\pi A}{\lambda} [C_1 e^{-\alpha z} + C_2 e^{\alpha z} + C_3 \cos(\beta z) + C_4 \sin(\beta z)] \cos\left(\frac{2\pi}{\lambda} y\right)\right\} \quad (\text{R2.21})$$

$$\psi = \arctan\left(\frac{\partial \delta}{\partial z}\right) = \arctan\left\{\frac{A}{2} [-\alpha C_1 e^{-\alpha z} + \alpha C_2 e^{\alpha z} - \beta C_3 \sin(\beta z) + \beta C_4 \cos(\beta z)] \sin\left(\frac{2\pi}{\lambda} y\right)\right\} \quad (\text{R2.22})$$

There is a uniform pulse magnetic field applied to the buckled strip. Thus, $\mathbf{M}' = M_0[1 \ 0 \ 0]^T$, where M_0 represents the magnitude of remanent magnetization. By combining Eqs. R2.19 and R2.20, we can obtain the expression of \mathbf{M} , as shown in Eq. R2.23.

$$\mathbf{M} = \begin{bmatrix} M_x \\ M_y \\ M_z \end{bmatrix} = M_0 \begin{bmatrix} \cos(\psi)\cos(\theta) \\ \sin(\theta) \\ -\sin(\psi)\cos(\theta) \end{bmatrix} \quad (\text{R2.23})$$

To validate the theoretical model, we employed a strip with $L = 28$ mm, $H = 2.7$ mm, $T = 0.5$ mm. The variation of magnetization vector along the y direction at $z=0$ and $z=H$ is shown in

Figure R2.7(a). The magneto-optical result in Figure R2.7(b) demonstrates the distribution of the magnetic flux density which is consistent with the simulation result based on the calculated magnetization profile.

Figure R2.7. Magnetization profile of the strip and validation of magnetic field distribution. **a** Magnetization profile of the strip at $z=0$ and $z=H$. **b** Validation of magnetic field distribution. The experimental result is obtained by a magneto-optical sensor and the modelling result is obtained based on the calculated magnetization profile.

R2.3: Magnetic response behavior

Figure R2.8. Schematic diagram of the magnetic actuation process.

To elucidate the impact of external magnetic field on the deformation of the magnetic elastomer, a theoretical model to illustrate the relationship between magnetic torque and out-of-plane displacement (δ_m) was established. The out-of-plane displacement of the strip show in Figure R2.8 can be calculated using plate theory for thin plate, as shown in Eqs. R2.24 and R2.25.

$$M_z = D \left(\frac{\partial^2 \delta_m}{\partial z^2} + \nu \frac{\partial^2 \delta_m}{\partial y^2} \right) = \iiint_V (-\tau_m) dV \quad (\text{R2.24})$$

$$\tau_m = [0 \quad 0 \quad 1] \mathbf{R}' \mathbf{M} \times \mathbf{B} \quad (\text{R2.25})$$

τ_m is the magnetic torque and M_z represents the bending moment applied to the elastomer. \mathbf{R}' is the rotational matrix for the change of magnetization direction during the deformation. V represents the volume of the plate. To estimate the magnetic actuated deformation, we utilize small deflection assumption by approximating \mathbf{R}' as an identity matrix, and assume that the effective body torque distribution along the elastomer strip is equivalent to that generated by external bending moment applied at the free end of the strip [Zhao, Ruike, et al. "Mechanics of hard-magnetic soft materials." *Journal of the Mechanics and Physics of Solids* 124 (2019): 244-263.]. Furthermore, we express the out-of-plane displacement as $\delta_m(y, z) = \theta_m(y) \times \Lambda_m(z)$. With these simplifications, Eq. R2.26 can be obtained for the case that an external magnetic field is applied along the x direction with magnitude of B_0 .

$$D \left(\theta_m \frac{d^2 \Lambda_m}{dz^2} + \nu \Lambda_m \frac{d^2 \theta_m}{dy^2} \right) = \int_0^y \int_0^H \{M_0 B_0 T \times \sin[\theta(y, z)]\} dy dz \quad (\text{R2.26})$$

The function $\sin[\theta(y, z)]$ can be fitted as: $\sin[\theta(y, z)] \approx G(z) \times \cos(ky + \phi)$. Based on the right side of Eq. R2.26, we can define θ_m as: $\theta_m = \sin(ky + \phi)$. Thus, Eq. R2.26 can be written as:

$$D \left(\frac{d^2 \Lambda_m}{dz^2} - \nu k^2 \Lambda_m \right) \sin(ky + \phi) = \int_0^H \left\{ \frac{M_0 B_0 T}{k} \times G(z) \right\} dz \times \sin(ky + \phi) \quad (\text{R2.27})$$

By solving Eq. R2.27 with the boundary condition ($\Lambda_m(z = 0) = 0$) numerically, we could obtain the result of $\delta_m = \Lambda_m \times \sin(ky + \phi)$. We employed a strip with $L = 28$ mm, $H = 4.3$ mm, $T = 0.5$ mm to validate the model, as shown in Figure R2.9. There is an agreement with the experimental result of the deformed strip.

Figure R2.9. Comparison of experimental results and theoretical model for the deformation of the strip under magnetic actuation.

In summary, we perform theoretical studies for the solvent and magnetic responsive behaviors of the magneto-elastomer. The theoretical models could give the quantitative relationship between the key parameters of buckling transformation and geometric parameters, the possible deformation modes of buckling structures, the quantitative description of 3D heterogeneous magnetization profiles, and the deformation under magnetic stimuli. The studies provide a theoretical guideline for the structural design and understanding of the underlying transformation mechanisms.

Comment 3. The two functions demonstrated, the transport and merging of droplets and the selective trapping, directly involve the interaction between objects being manipulated and

underlying microstructures. However, the fluorescent signal enhancement does not appear to involve direct interaction between the aerosol droplets and the microstructure, so one would ask, why do we need this particular type of dynamic transformation to cause the disturbance that enables the aerosol droplet merging? The authors need to think about the unique properties and functions of the dynamic transformations of their microstructure and explore how they can be used.

Response: Thank you for bringing up this point.

Firstly, the aggregation of aerosol droplets is achieved via the flow field generated by the dynamic transformation of elastomer structure, whose mechanism is indeed similar with the droplets transport and merging experiments. The transformations of the elastomers could impose a variety of impacts on the fluid environment. Through the PIV analysis shown in Figure 6 of the revised manuscript, it can be found that the dynamic transformations of the elastomer structures do not disturb the fluid environment arbitrarily, as they can realize the directional transfer and mixing of the liquid. Thus, the dynamic elastomer structures could be used to achieve the aggregation of aerosol droplets through the dynamic flow field with merging capability and do not require the direct contact with the droplets.

Secondly, the energy consumption and efficiency of fluidic manipulation is analyzed based on the proposed unique transformation in this work, which is verified as a relatively efficient fluidic manipulation method compared with those reported by other literatures (Discussion R3). Therefore, the dynamic transformation in buckling structure can achieve the efficient aggregation of aerosol droplets and has good application prospects in detection analysis and microfluidic devices.

Thirdly, to further demonstrate the potential function of the method, we apply the proposed dynamic transformation to a microfluidic device, which can achieve efficient mixing of viscous liquids at a low Reynolds number environment as well as in a dynamic environment, as shown in Discussion R4.

Finally, the dynamic transformation of elastomer structure generates sufficient output forces and can be used as an effective propulsion strategy. Therefore, a wireless swimming soft robot actuated by programmable magnetic field input is fabricated, and the dynamic transformation of the buckling structure can help to realize the controllable navigation of the robot, as shown in the Discussion R5.

All of the results presented afterwards try to fully exploit the unique properties and functions of the dynamic transformations in this work for different applications, and we hope they can address your comments at a satisfactory level. According to your comment, we have added the following discussion in the “**2.5 Flow field induced by the morphological transformation under dynamic magnetic field**” part and “**2.6 Potential applications of the dynamic transformation of non-freestanding structures**” part of the revised manuscript.

Discussion R3: In this part, we calculate the energy consumption of the developed fluidic manipulation method, and perform the comparison of the pumping performance reported by other literatures.

To evaluate the pumping efficiency of fluid generation methods, a dimensionless parameter (ε_{ef}) similar to references [1. Eloy, Christophe, and Eric Lauga. "Kinematics of the most efficient cilium." *Physical Review Letters* 109.3 (2012): 038101; 2. Osterman, Natan, and Andrej Vilfan. "Finding the ciliary beating pattern with optimal efficiency." *Proceedings of the National Academy of Sciences* 108.38 (2011): 15727-15732.] is adopted as Eq. R2.28.

$$\varepsilon_{ef} = \frac{\eta_{vis} Q^2}{V_{vol} P} \quad (\text{R2.28})$$

η_{vis} and Q represent the viscosity of the pumping fluid and volume flow rate, respectively. V_{vol} is the volume of the magnetic elastomer used for pumping fluid and P is the input power. The volume flow rate $Q = 25.6 \mu\text{l}/\text{min}$ of the proposed method is obtained by measuring the change in mass of the fluid flowing from the outlet with the actuation frequency of 5 Hz [3. Zhou, Mingxing, et al. "Miniaturized soft centrifugal pumps with magnetic levitation for fluid handling." *Science Advances* 7.44 (2021): eabi7203.]. Glycerol is adopted as the fluid with a viscosity η_{vis} of $0.876 \text{ N} \cdot \text{s}/\text{m}^2$. The input power P is determined by integrating the work performed by magnetic torque on the dynamic transformation of the magnetic elastomer [4. Ren, Ziyu, et al. "Multi-functional soft-bodied jellyfish-like swimming." *Nature Communications* 10.1 (2019): 2703.], as shown in Eq. R2.29, where $\omega(x, y, z, t)$ represent angular velocity and T is the time for one actuation cycle.

$$P = \frac{1}{T} \int_0^T \int_{V_{vol}} |(\mathbf{R}'\mathbf{M} \times \mathbf{B}) \cdot \omega| dV dt \quad (\text{R2.29})$$

Eq. R2.30 is used to calculate the kinematic property of the magnetic elastomer over one actuation cycle.

$$\delta_m(x, y, z, t) = \Lambda_m(z) \times \sin\left(\frac{2\pi}{\lambda} y + \phi(t)\right) \quad (\text{R2.30})$$

Λ_m is the magnitude of the out-of-plane displacement under magnetic actuation and $\phi(t)$ is the phase shift induced by the varying magnetic field direction and can be measured from experimental results. According to the analysis of magnetic response behavior, the rotational deflection and angular velocity can be obtained as Eqs. R2.31 and R2.32.

$$\alpha = \arctan\left(\frac{\partial \delta_m}{\partial y}\right) = \arctan\left\{\frac{2\pi \Lambda_m}{\lambda} \cos\left[\frac{2\pi}{\lambda} y + \phi(t)\right]\right\} \quad (\text{R2.31})$$

$$\omega(x, y, z, t) = \frac{\partial \alpha}{\partial t} = \frac{-2\pi \Lambda_m}{\lambda \left[1 + \left(\frac{\partial \delta_m}{\partial y}\right)^2\right]} \cdot \sin\left(\frac{2\pi}{\lambda} y + \phi(t)\right) \cdot \frac{d\phi(t)}{dt} \quad (\text{R2.32})$$

By applying a rotating magnetic field $\mathbf{B} = B_0 \left[\cos\left(\frac{2\pi t}{T}\right) \quad \sin\left(\frac{2\pi t}{T}\right) \quad 0 \right]^T$, the input energy can be calculated as Eq. R2.33.

$$P = \frac{1}{T} \int_0^T \int_{V_{vol}} \left| M_0 B_0 \cdot \sin\left(\frac{2\pi t}{T} - \alpha - \theta\right) \cdot \omega \right| dV dt \quad (\text{R2.33})$$

By solving Eq. R2.33 numerically, we could obtain the average input power as 1.4×10^{-3} W. The power done by magnetic torque contributes to the storage of strain energy, the variation of kinetic energy of the magnetic elastomer and fluid.

The comparison of the pumping efficiency ϵ_{ef} with other methods is shown in Table R2.1, which demonstrates that the developed flow generation method outperforms most reported soft pumping device in terms of pumping efficiency.

Table R2.1. Comparison of the pumping efficiency with literature

Refs	Actuation strategy	f (Hz)	Medium	Re	Q (ul/min)	P (W)	V (mm ³)	ϵ_{ef}
1	Magnetic actuation shape-memory pump	270	Glycerol solution	NA	1800	0.208	25.0	92.9
2	Magnetically actuated peristaltic motion	20	Water	NA	74.7	0.016	78.4	0.07
3	Chamber deformation via electromagnetic actuation	1~20	Water	NA	67.5	0.049	24.7	0.06
4	Cilium motion	31.8	Water	NA	2.1×10^{14}	2.5×10^{14}	5.4×10^{10}	30.4
5	Cilia array inspired motion	2.5	Glycerol solution	0.03	150	NA	NA	NA
6	Magnetically actuated centrifugal pump	1.6~16.6	Water and fat emulsion	NA	30600	4.1	NA	NA
This work	Magnetically actuated dynamic transformation	5	Glycerol solution	0.004	25.6	0.0014	25.9	262.6

Note: f : Actuation frequency; Q : Volume flow rate; P : Power; V : Volume of the actuator; ε_{ef} : Pumping efficiency (Dimensionless). [1. Smith, Aaron R., et al. "Characterization of a high-resolution solid-state micropump that can be integrated into microfluidic systems." *Microfluidics and Nanofluidics* 18.5 (2015): 1255-1263; 2. Saren, A., A. R. Smith, and K. Ullakko. "Integratable magnetic shape memory micropump for high-pressure, precision microfluidic applications." *Microfluidics and Nanofluidics* 22.4 (2018): 38; 3. Rusli, M. Q. A., et al. "Electromagnetic actuation dual-chamber bidirectional flow micropump." *Sensors and Actuators A: Physical* 282 (2018): 17-27; 4. Eloy, Christophe, and Eric Lauga. "Kinematics of the most efficient cilium." *Physical Review Letters* 109.3 (2012): 038101; 5. Dong, Xiaoguang, et al. "Bioinspired cilia arrays with programmable nonreciprocal motion and metachronal coordination." *Science Advances* 6.45 (2020): eabc9323; 6. Zhou, Mingxing, et al. "Miniaturized soft centrifugal pumps with magnetic levitation for fluid handling." *Science Advances* 7.44 (2021): eabi7203.]

Discussion R4: In this part, the dynamic transformation of buckling structure is used for efficient mixing of viscous fluids in microfluidic devices at low Reynolds number.

Figure R2.11. Fluid mixing at low Re using the dynamic transformation of the magneto-elastomers. **a** Schematic diagram of the microfluidic device with Y-shaped fluidic channel. **b** Images of the fluid mixing device including glycerol layers with two colors and magneto-elastomers. **c** Dynamic transformation of the magneto-elastomers in the fluidic channel under rotating magnetic field. **d** Efficient fluid mixing with 0 flow rate of the inlet. **e** Efficient fluid mixing with 40 mL/h flow rate of the inlet. The scale bars are 3 mm.

The mixing of fluids in small-scale channels is important for the development and application of microfluidic devices. Nevertheless, efficient mixing is challenging due to the low Reynolds number (Re) and slow diffusion effects. Existing methods include passive fluid mixing devices, which often require specific channel designs, such as spiral microchannels, or straight channels

with groove or ridge microstructures [1. Stroock, Abraham D., et al. "Chaotic mixer for microchannels." *Science* 295.5555 (2002): 647-651; 2. Therriault, Daniel, Scott R. White, and Jennifer A. Lewis. "Chaotic mixing in three-dimensional microvascular networks fabricated by direct-write assembly." *Nature Materials* 2.4 (2003): 265-271; 3. Zhao, Qianbin, et al. "A review of secondary flow in inertial microfluidics." *Micromachines* 11.5 (2020): 461.]. In addition, active mixing devices have also been developed, including magnetically actuated artificial cilia and bubble-based microfluidic devices, which may have limited mixing thoroughness or need special design of ciliary arrays to achieve optimized metachronal wave [4. Dong, Xiaoguang, et al. "Bioinspired cilia arrays with programmable nonreciprocal motion and metachronal coordination." *Science Advances* 6.45 (2020): eabc9323; 5. Khoshmanesh, Khashayar, et al. "A multi-functional bubble-based microfluidic system." *Scientific Reports* 5.1 (2015): 9942.]. Here, we apply magneto-elastomers with dynamic structural transformations for fluid mixing at low Re in microfluidic devices, as shown in Figure R2.11. Glycerol colored with different dyes is used as the fluid at the inlet and fills the entire microfluidic channel. The volume of the straight channel is 182 mm^3 , and no apparent diffusion of fluid occurs in the channel under static state. Two magnetic strips with a height of 1.6 mm are used for the fluid mixing. With a dynamic magnetic field (actuation frequency: 5 Hz), we demonstrate that this mixing device could rapidly mix viscous glycerol with local Re less than 0.004. In addition, we also verify the efficient fluid mixing under external dynamic fluid environment (flow rate: 40 mL/h).

Discussion R5: In this part, the dynamic transformation of buckling structure is further applied for the efficient and controllable propulsion of untethered swimming robots.

Figure R2.12. Untethered swimming robots actuated by the dynamic geometric transformation of magneto-elastomers. **a** Schematic diagram of the propulsion of the swimming robot at the

air-water interface and the robotic structure. **b** Top view and side view of the swimming robot during the actuation to fulfill a rectangular trajectory. **c** Side view of the deformation of the robotic structure during the actuation. The scale bars are 5 mm.

Underwater creatures will generate propulsion or achieve movement stability and controllability by controlling wave-like fins, which provides inspiration for the development of biomimetic swimming robots [1. Sefati, Shahin, et al. "Mutually opposing forces during locomotion can eliminate the tradeoff between maneuverability and stability." *Proceedings of the National Academy of Sciences* 110.47 (2013): 18798-18803; 2. Ruiz-Torres, Ricardo, et al. "Kinematics of the ribbon fin in hovering and swimming of the electric ghost knifefish." *Journal of Experimental Biology* 216.5 (2013): 823-834.]. Ribbon fin-based propulsions have been achieved at the centimeter scale (20~50 cm) by using electromechanical systems to control the locomotion of multiple nodes on the ribbon [3. Liu, Hanlin, Bevan Taylor, and Oscar M. Curet. "Fin ray stiffness and fin morphology control ribbon-fin-based propulsion." *Soft Robotics* 4.2 (2017): 103-116; 4. Arslan, Erdem, and A. K. Ç. A. Kadir. "A design methodology for cuttlefish shaped amphibious robot." *Avrupa Bilim ve Teknoloji Dergisi* (2019): 214-224; 5. Shen, Qi, et al. "Basic design of a biomimetic underwater soft robot with switchable swimming modes and programmable artificial muscles." *Smart Materials and Structures* 29.3 (2020): 035038;]. But this strategy is difficult to build smaller-scale untethered swimming robots due to the limited on-board space. In comparison, magnetic control method exhibits high controllability and programmability for small-scale soft robots.

In this work, by utilizing the proposed buckling instability-encoded heterogeneous magnetization profiles and programmable magnetic field inputs, an untethered swimming robot is developed to achieve effective swimming behavior with excellent controllability. We adopt a polylactic acid flat plate as the substrate (20 mm in length), and coat a thin layer of silicone elastomer on its surface to facilitate the robust bonding between the substrate and the magneto-elastomers (16 mm in length and 5 mm in height), as shown in Figure R2.12(a). An oscillating magnetic field generated by three-axis Helmholtz coils is applied to actuate the dynamic transformation of the magneto-elastomers and achieve the efficient propulsion of the swimming robot at the air-water interface. The input magnetic field strength and frequency are 9 mT and 12 Hz, respectively. By continuously adjusting the direction angle of the magnetic field, the robot could achieve omnidirectional locomotion and move along a rectangular trajectory, as shown in Figure R2.12(b) and the supplementary video. Further increasing the strength of the output magnetic field could improve the magnetic control capability and contribute to the miniaturization of the proposed swimming robot.

There are a few more minor issues:

1. The methods section needs to include more detailed experimental procedures for each of their experiments shown in the figures and supplementary figures. At the moment, they are too general.

Response: Thank you for the comment. We supplement detailed experimental procedures in the “Experimental section” of the revised manuscript as below.

Preparation of magneto slurry: The magnetic elastomers were prepared by blending silicone-based materials (Ecoflex 00-30 part A and part B) and NdFeB magnetic microparticles in a 1:1 mass ratio. After thoroughly mixing, vacuum degassing was performed to remove air bubbles for 1 min in a vacuum chamber.

Fabrication of 3D non-freestanding structures:

(i) Fabrication of surface-attached strip structures: First, various groove patterns (e.g., rectangular holes) were cut on the acrylic sheets by a laser processing platform (Universal Laser Systems PLS6.75). Hydrophobic treatment was performed for the surfaces of acrylic sheets with groove patterns: Acrylic molds were treated with plasma for 10 min following with silane steam treatment for 3 h using triethoxy(1H,1H,2H,2H-perfluoro-1-octyl)silane (POTS). A glass substrate with hydrophilic surface obtained by plasma treatment for 10 min was prepared. The acrylic mold and glass substrate are held together by clips. The uncured magneto-elastomer was poured onto the surface of the acrylic mold, and the entire structures were placed in a vacuum chamber for 5 min so that the elastomer could thoroughly fill the cavity portion of the acrylic mold. After degassing, the magneto-elastomer was cured in an oven (60 °C) for 20 min. The elastomer structure was peeled off from the acrylic mold and stayed on the glass substrate and the strong adhesion between the elastomer structure and the glass substrate allows the formation of stable non-freestanding structures. To endow the structure with fluorescent effect, a thin layer of silicone elastomer doped with fluorescence powder (weight ratio 1 : 0.08) was added on top of the elastomeric structure.

(ii) Fabrication of surface-attached cellular structures: Laser cutting was performed to form lattice-shaped acrylic sheets by the laser processing platform. The lattice-shaped acrylic sheets were placed in a container and PDMS prepolymer (prepared by mixing base and curing agent of sylgard 184 with a weight ratio 10:1) was poured into the container. The PDMS polymer was cured in an oven (60 °C) for 1 h and after that the acrylic sheets were peeled off to form PDMS-based mold with lattice-shaped groove. Hydrophobic treatment was performed for the PDMS-based mold: the PDMS mold was treated with plasma for 10 min following with silane steam treatment for 3 h using triethoxy(1H,1H,2H,2H-perfluoro-1-octyl)silane (POTS). After the silane treatment, the PDMS surface was cleaned via ethanol and dried in the oven (60 °C) for 1 h. The magnetic slurry was poured onto the surface of the PDMS mold, and were placed in a vacuum chamber for 5 min so that the slurry could thoroughly fill the cavity portion. A glass substrate with hydrophilic surface obtained by plasma treatment for 10 min was covered on top of the magnetic slurry which was cured in the oven (60 °C) for 20 min. The cellular elastomer structure was released from the acrylic mold and stayed on the glass substrate.

(iii) Fabrication of surface-attached micro cellular structures: First, a resin mold ($9\times 9\times 1.4$ mm³) with lattice-shaped microgrooves was fabricated by a 3D printer (NanoArch S130, BMF Precision, China) and poly(ethylene glycol) diacrylate (PEGDA) resin. To avoid the residual

resin affecting the curing of the elastomer, the resin mold was irradiated by UV light for 1 hour, and cleaned via ethanol, and followed by the hydrophobic treatment. The magnetic slurry fills the microgrooves of the resin mold in a vacuum chamber and is covered by a hydrophilic-treated glass substrate. After the magnetic elastomer was cured, the cellular microstructure was peeled off from the resin mold.

PIV analysis:

The liquid environment for the PIV analysis was formed by glycerol and fluorescence powders were used as tracer. The distance between the air-water interface and the upper surface of the elastomer structure was kept at 3 mm. Hirox RH-2000 Digital Microscope was used to record the process. The flow field was obtained by performing PIV analysis using an open-source package MatPIV 1.7 (<https://www.mn.uio.no/math/english/people/aca/jks/matpiv/>).

Droplet manipulation via the transformation of the strip structures:

The ethanol solution in which Rhodamine B was dissolved was used as a fluorescent droplet. As shown in Figure 4, the fluorescent droplets distributed on the upper layer of the container under UV light irradiation excited red light. The two fluorescent droplets in Figure 4e have different Rhodamine B contents (mass ratio of 0.02% and 0.075%) to make them exhibit different fluorescence intensities. The viscous liquid environment is formed by silicone oil (~200 mpa.s). The dynamic magnetic field is generated by a rotating sphere magnet with a diameter of 35 mm, as shown in Figure S10, and the surface magnetic field strength of the magnetic sphere is 200 mT. The rotational frequency of the sphere magnet during the manipulation process is 5 Hz. The distance between the upper surface of the sphere magnet and the elastomer structure is about 4 mm. During the actuation process, the locomotion of the droplet is controlled by adjusting the rotation direction and position of the sphere magnet.

In-situ particle manipulation:

Particle manipulation was performed underwater. Air bubbles inside the elastomer structure were removed by placing it in a vacuum chamber for 5 min. 200 and 600 μm particles in Fig. 5(b) were fabricated by projection micro stereolithography of PEGDA resin. 200 and 400 μm fluorescent particles in Fig. S25 were prepared by 3D printing of photocurable resin composed of PEGDA, isobornyl acrylate, and green or red fluorescent powders with a mass ratio of 1:9:0.05. The lattice microstructure is actuated by rotating sphere magnet at a frequency of 2 Hz. The distance between the elastomer structure on the upper surface of the sphere magnet is about 4 mm.

Collection of sprayed aerosol:

The aerosol is prepared by the ethanol solution doped with fluorescein isothiocyanate with a mass ratio of 0.13%. The microdroplets were generated by a sprayer. The viscous liquid environment is formed by silicone oil. The sphere magnet was placed beneath the strip

structures and actuated at a frequency of 5 Hz to collect these microdroplets. The brightness of the fluorescence signal was calculated as $Bright = 0.299*R+0.587*G+0.114*B$, where R , G , and B refer to the RGB value of each pixel.

Fabrication and actuation of untethered swimming robots:

A polylactic acid (PLA) flat plate fabricated by fused filament fabrication was used as substrate. To ensure a stable connection between the substrate and magneto-elastomer, the flat plate was coated by a thin layer of Ecoflex elastomer. An acrylic sheet with groove pattern was fabricated by a laser processing platform and placed on top of the flat plate. The magnetic slurry was poured onto the surface of the acrylic mold, and the entire structures were placed in a vacuum chamber for 5 min so that the elastomer could thoroughly fill the cavity portion of the acrylic mold. After degassing, the magneto-elastomer was cured in an oven (60 °C) for 20 min. The elastomer structure was peeled off from the acrylic mold and stayed on the PLA substrate. The whole structure was placed in toluene to induce the buckling transformation and are magnetized via the magnetizer. Two magnetized elastomer structures are connected together as the robotic structure via uncured silicone elastomer as the bonding agent. The untethered swimming robot was actuated by a three-axis Helmholtz coils which generate an oscillating magnetic field with the magnetic field strength and frequency of 9 mT and 12 Hz, respectively, and were controlled by a custom-programmed software with LABVIEW.

Fluidic mixing device:

A PDMS device with Y-shape channel was fabricated. The PDMS device and the glass substrate with the elastomer strips are treated by plasma for 10 min. The PDMS device is placed on top of the glass substrate by pressing for 2 h to form stable connection. Glycerol colored with two different dyes (mass ratio: 0.01%) was pumped into the fluidic channel via peristaltic pump and formed a stratified fluid. The magnetic strips were actuated by the sphere magnet beneath the device at a frequency of 5 Hz to form turbulent flow under low Reynolds number environment.

Simulation methods:

(i) The simulations of magnetic fields generated by cuboid permanent magnet and spherical magnet were performed by COMSOL. (ii) The simulation of the buckling transformation of magneto-elastomers in solvent was performed by ABAQUS. The simulations include the parameters: the elastic modulus of the elastomer (67 kPa) and the swelling ratio. The non-freestanding structures in the simulation are assumed to be thermally expanded. The buckling transformations of the structures are simulated by setting the change of environment temperature and the coefficient of thermal expansion so that they undergo the same volume expansion as the swelling process in solvent. (iii) The simulation of the transformation under magnetic stimuli was conducted by a user-element subroutine (UEL) in ABAQUS. Considering the asymmetry of Cauchy stress induced by magnetic body torques, a customized 8-noded brick elements are employed to handle the nine components of the Cauchy stress

tensor. (iv) The simulation of the coupling stimulation of magnetic field and solvent was performed by COMSOL. Elastomer structures were divided into smaller subsections with varying magnetic torques in COMSOL structural mechanics module. The torques of subsections were recalculated according to the deformation of the elastomer structure until it reaches the equilibrium state. Subsequently, solvent stimulation was applied. The deformed elastomer structure could undergo the swelling process and buckling transformation which are simulated through thermal expansion function.

Characterization Techniques:

The magnetization profile was measured by a magneto-optical sensor (MagViewS, Matesy, Jena, Germany). The mechanical property of the magneto-elastomers was measured by a mechanical tester (MACH-1 mechanical tester v500cst, MA008). The magnetic hysteresis of the magnetic elastomer was measured by a PPMS model 6000 Quantum Design VSM.

2. Fig. 3a, calling these different connectivity types “boundary conditions” seems to be another misnomer.

Response: Thanks for your comment. We have changed the term “boundary conditions” to “connectivity types” in the revised manuscript and supporting information.

3. Line 133 – 134, the microscopic swelling of silicone elastomer, the infusion of solvent in polymer chains, should occur throughout the structure, even though the extent of the swelling may not be observable by light microscopy.

Response: Thank you for the comment. We agree with your view and have revised the description of the swelling and buckling process of the surface-attached elastomeric structure in the “Results and discussion” section of the main text. When the magneto-elastomer is immersed in toluene solution, swelling behavior of elastomer structure is induced by the diffusion of toluene into polymer matrix. While the swelling process of bottom part would be affected by the constraint of substrate, axial compressive forces are formed to provide the buckling condition of the swelled elastomer. According to your suggestion, we have revised the description in the “**2.1 Shape-morphing mechanisms of the magneto-elastomers**” part of the revised manuscript.

Response to Reviewer #3

The manuscript presents a very interesting way to fabricate dynamically reconfigurable structures employing buckling and magnetic actuation. Specifically, a non-freestanding composite structure of elastomer with magnetic particles. The structure is firstly buckled by solvent swelling. While maintaining the buckled state, an external magnetic field is applied to 'imprint' the magnetic flux orientation of those embedded magnetic particles. After de-swelling, the structure returns to its initial flat shape. And this shape can be changed into various shapes by applying an external magnetic field, thanks to the imprinted magnetic flux in magnetic particles. Using this principle, the authors have demonstrated the generation of fluid flow by dynamic modulation of the shape with an external magnetic field. And, selective particle trapping and sensitive droplet detection have also been shown using the suggested structures.

Overall, the manuscript seems quite interesting, but following questions/suggestions should be considered while preparing their revised version of the manuscript.

Response: Thank you very much for your positive feedback. Based on your comments, we have made corresponding discussions and changes to improve the manuscript.

1. First of all, I don't find any explanation on the mechanism and theoretical analysis on the observed phenomenon. The authors have shown FEM results only, without discussing involved mechanics and magnetomechanical coupling.

Response: Thanks for your kind suggestion. According to your suggestion, theoretical analysis and mechanism explanation is added in the revised manuscript. We supplement the analytical modeling of this work including (1) swelling-induced buckling behaviors, (2) buckling encoded 3D heterogeneous magnetization profile, and (3) magnetic response behavior. The following discussion is added in the in the “**2.2 Characterization of magnetic and solvent responsive behavior of surface-attached structures**” part and “**2.4 Morphological transformations of cellular structures**” part of the revised manuscript and “**Supplementary Note 1 Analytical model**” part of the revised supporting information.

(1) Swelling-induced buckling behaviors:

The swelling properties of silicone elastomers to solvents are related to the Hildebrand solubility parameter and the polarity of the solvent [I. Rumens, C. V., et al. "Swelling of PDMS networks in solvent vapours; applications for passive RFID wireless sensors." *Journal of Materials Chemistry C* 3.39 (2015): 10091-10098.]. Since many non-polar solvents such as toluene and n-Hexane have high affinity for elastomers, upon contact, the non-polar solvents

migrate through the elastomer by diffusion and cause the swelling of the elastomer structure [2. Lee, Jessamine Ng, Cheolmin Park, and George M. Whitesides. "Solvent compatibility of poly (dimethylsiloxane)-based microfluidic devices." *Analytical Chemistry* 75.23 (2003): 6544-6554; 3. Hu, Yuhang, et al. "Indentation of polydimethylsiloxane submerged in organic solvents." *Journal of Materials Research* 26.6 (2011): 785-795; 4. Kim, Yongjin, Jay van den Berg, and Alfred J. Crosby. "Autonomous snapping and jumping polymer gels." *Nature Materials* 20.12 (2021): 1695-1701.]. Some polar solvents such as water and ethanol do not cause swelling of silicone elastomers. By removing the absorbed solvent, the silicone elastomer could be restored to its original shape, and ethanol is chosen as the solvent for the deswelling of the elastomer due to the high miscibility of toluene with ethanol.

Figure R3.1. Schematic diagram of buckling transformation. **a** Transformation of elastomer strip. **b** Transformation of elastic plate with free-to-rotate junctions.

A thin elastomer strip has length L , height H , and thickness T as shown in Figure R3.1 (a). The bottom ($z=0$) and top ($z=H$) edge of the elastomer strip are fixed and free, respectively. Foppl-von Karman equations were used to describe the deformation of thin flat plates [5. Jiang, Ruiqi, Jianliang Xiao, and Jizhou Song. "Buckling of thin gel strip under swelling." *Theoretical and Applied Mechanics Letters* 7.3 (2017): 134-137; 6. Kang, Sung Hoon, et al. "Buckling-induced reversible symmetry breaking and amplification of chirality using supported cellular structures." *Advanced Materials* 25.24 (2013): 3380-3385.], as shown in Eqs. (R3.1) and (R3.2).

$$\delta(y, z) = \frac{A}{2} \times f(z) \times \sin\left(\frac{2\pi}{\lambda} y\right) \quad (\text{R3.1})$$

$$\frac{\partial^4 \delta}{\partial y^4} + 2 \frac{\partial^4 \delta}{\partial y^2 \partial z^2} + \frac{\partial^4 \delta}{\partial z^4} + \frac{t_{11}}{D} \frac{\partial^2 \delta}{\partial y^2} = 0 \quad (\text{R3.2})$$

$\delta(y, z)$ represents the out-of-plane displacement of the strip. $u_y(y, z)$ and $u_z(y, z)$ are the in-plane displacement along the y and z direction, respectively. λ represents the wavelength. t_{11} and D are the membrane force along the y direction and bending stiffness, respectively. A is the peak-to-peak value. $f(z)$ represents a normalized function, where $f(H) = 1$. By solving the ordinary differential equation for $f(z)$, a general solution can be obtained as Eq. (R3.3).

$$f(z) = C_1 e^{-\alpha z} + C_2 e^{\alpha z} + C_3 \cos(\beta z) + C_4 \sin(\beta z) \quad (\text{R3.3})$$

$$\alpha = \sqrt{\sqrt{\frac{t_{11}}{D} \left(\frac{2\pi}{\lambda}\right)^2} + \left(\frac{2\pi}{\lambda}\right)^2} \quad (\text{R3.4})$$

$$\beta = \sqrt{\sqrt{\frac{t_{11}}{D} \left(\frac{2\pi}{\lambda}\right)^2} - \left(\frac{2\pi}{\lambda}\right)^2} \quad (\text{R3.5})$$

The values of C_1, C_2, C_3, C_4 can be determined by the boundary conditions shown in Figure R3.1. For the strip structure in Figure R3.1(a), the clamped edge and the free edge of the strip determine the equations:

$$\left\{ \begin{array}{l} f(0) = 0 \\ \frac{\partial f}{\partial z} \Big|_{(z=0)} = 0 \\ \left(\frac{\partial^2 \delta}{\partial z^2} + \nu \frac{\partial^2 \delta}{\partial y^2} \right) \Big|_{(z=H)} = 0 \\ \left[\frac{\partial^3 \delta}{\partial z^3} + (2 - \nu) \frac{\partial^3 \delta}{\partial y^2 \partial z} \right] \Big|_{(z=H)} = 0 \end{array} \right. \quad (\text{R3.6})$$

where ν is the Poisson's ratio. For the strip structure in Figure R3.1(b), the free-to-rotate junctions determine the equations.

$$\left\{ \begin{array}{l} \delta(0, z) = \delta(L, z) = 0 \\ \left(\frac{\partial^2 \delta}{\partial y^2} + \nu \frac{\partial^2 \delta}{\partial z^2} \right) \Big|_{(y=0)} = \left(\frac{\partial^2 \delta}{\partial y^2} + \nu \frac{\partial^2 \delta}{\partial z^2} \right) \Big|_{(y=L)} = 0 \end{array} \right. \quad (\text{R3.7})$$

With the boundary conditions shown in Eq. R3.6, we can get the equations for C_1, C_2, C_3, C_4 .

$$-\alpha C_1 + \alpha C_2 + \beta C_4 = 0 \quad (\text{R3.8})$$

$$C_1 + C_2 + C_3 = 0 \quad (\text{R3.9})$$

$$\mathbf{A}_{co} \begin{bmatrix} C_3 \\ C_4 \end{bmatrix} = \begin{bmatrix} A_{co11} & A_{co12} \\ A_{co21} & A_{co22} \end{bmatrix} \begin{bmatrix} C_3 \\ C_4 \end{bmatrix} = \begin{bmatrix} 0 \\ 0 \end{bmatrix} \quad (\text{R3.10})$$

$$\text{Det}(\mathbf{A}_{co}) = 0 \quad (\text{R3.11})$$

where

$$A_{co11} = \left(\beta^2 + \nu \frac{4\pi^2}{\lambda^2} \right) \cos(\beta H) + \left(\alpha^2 - \nu \frac{4\pi^2}{\lambda^2} \right) \cosh(\alpha H)$$

$$A_{co12} = \left(\beta^2 + \nu \frac{4\pi^2}{\lambda^2} \right) \sin(\beta H) + \beta \left(\alpha - \nu \frac{4\pi^2}{\alpha \lambda^2} \right) \cosh(\alpha H)$$

$$A_{co21} = \left(-\beta^3 + (2 - \nu) \beta \frac{4\pi^2}{\lambda^2} \right) \sin(\beta H) + \left(\alpha^3 - (2 - \nu) \alpha \frac{4\pi^2}{\lambda^2} \right) \sinh(\alpha H)$$

$$A_{co22} = \left(\beta^3 + (2 - \nu) \beta \frac{4\pi^2}{\lambda^2} \right) \cos(\beta H) + \left(\alpha^2 \beta - (2 - \nu) \beta \frac{4\pi^2}{\lambda^2} \right) \cosh(\alpha H)$$

As shown in Eq. R3.10, linear homogeneous equations determine the relationship between C_3 and C_4 . To obtain nonzero solution for the equations, the determinant of the coefficient matrix is equal to zero, i.e., $Det(A_{co})=0$, which gives the relationship between t_{11} and λ and determines the critical swelling condition.

Firstly, we analyze the case with free-to-rotate junctions (shown in Figure R3.1(b)). The relationship between λ and L can be determined by $W_n = L/\lambda_n$, where $2W_n$ is an integer and represents the deformation mode of strip with the junctions. By giving the value of W_n and solving Eq R3.11, we can calculate the critical force t_{11}^c , and the relationship between the normalized critical force ($\frac{H^2 t_{11}^c}{ET^3}$) and geometry parameters, as shown in Figure R3.2. In the theoretical study, we fix the values of H and T , and study the impact of aspect ratio (L/H) on the deformation mode W_n . The theoretical and experimental results show that the relationship between L/H and W_n is in a form of step function. When the L/H reaches a critical value, the mixed deformation modes would be exhibited as shown in Figure R3.2(b) ($L/H = 3.70$, $L/H = 7.41$). Mixed deformation mode results for other critical conditions can be verified by extensive swelling tests.

Figure R3.2. Impact of aspect ratio on the deformation mode of the strip with free-to-rotate junctions. **a** The relationship between the normalized critical force and aspect ratio under different deformation modes. Discrete dots in orange color represents the experimental results which can also be found in Fig. R3.2b. The theoretical results illustrate the deformation modes that the strip will exhibit when a critical force is reached during swelling for a given aspect ratio. The intersection of the two curves indicates that multiple deformation modes would occur with this geometry parameter. **b** Theoretical and experimental results for the relationship between the deformation mode (W_n) and the aspect ratio. The theoretical results show a relationship in the form of a step function, which indicates that when the aspect ratio is within a certain range, the deformation mode of the strip is fixed; when the aspect ratio reaches a critical value, the strip would exhibit mixed deformation modes. As the experimental results with $L/H = 3.70$, W_n can be equal to 1.0 or 1.5. When $L/H = 7.41$, W_n can be equal to 2.0 or 2.5. The scale bar is 2 mm.

Secondly, we analyze the case in Figure R3.1(a). By solving the Eq R3.11 numerically under given λ , we can calculate the value of t_{11} and gives the relationship between λ and t_{11} . We still use the parameter $W_n = L/\lambda_n$ to illustrate the deformation of the strip. While, in this case, W_n can be any positive real number. As shown in Figure R3.2(a), for a given W_n , there is a nonlinear relationship between L/H and normalized critical force. The critical deformation

mode $W_{nc} = L/\lambda_{nc}$ can be calculated according to the extreme point of the curve. Therefore, we could obtain the relationship between W_{nc} and L/H , as shown in Figure R3.3, which shows a linear correlation between W_{nc} and L/H . Through a linear fitting, λ_{nc} can be obtained, as shown in Eq. R3.15, which demonstrates that the length of the strip has negligible effect on the wavelength of the buckled strip. With the value of λ and t_{11} , the results of parameters including C_1 , C_2 , C_3 , C_4 , α , and β can be obtained.

$$\lambda_{nc} \approx 3.256H \quad (\text{R3.15})$$

The amplitude of the out-of-plane displacement (δ) can be obtained via the minimization of the total energy [5. Jiang, Ruiqi, Jianliang Xiao, and Jizhou Song. "Buckling of thin gel strip under swelling." *Theoretical and Applied Mechanics Letters* 7.3 (2017): 134-137.], as shown in Eqs. R3.16 and R3.17.

$$U = \frac{0.05ET^3A^2}{H^2(1-\nu^2)} - \frac{0.0434ETA^2(\chi-1)}{\chi H(1-\nu^2)} + \frac{0.3745ETH(\chi-1)^2}{\chi^2(1-\nu^2)} + \frac{0.0029ETA^4}{H^3(1-\nu^2)} \quad (\text{R3.16})$$

$$\frac{\partial U}{\partial A} = 0 \quad (\text{R3.17})$$

U represents the total energy of the strip and χ represents the swelling ratio. The values of the peak-to-peak amplitude (A) can be calculated.

$$A = 2.724H \sqrt{\frac{\chi-1}{\chi} - 1.155 \frac{T^2}{H^2}} \quad (\text{R3.18})$$

Experimental results were employed to validate the theoretical model of the buckled strip, as shown in Figure. R3.3. The agreement between the experimental and theoretical results validates the model which could quantitatively describes the relationship between the buckling results and the geometry parameters.

Figure R3.3. Theoretical and Experimental results for the deformation of the buckled strip. **a** Theoretical results and fitting for the relationship between W_{nc} and the aspect ratio (L/H). **b**

The impact of the height of the strip on the wavelength. **c** The impact of the height of the strip on the amplitude. **d** Comparison of the deformed shape of the strip structure.

(2) Buckling encoded 3D heterogeneous magnetization profile:

Figure R3.4. Schematic diagram of the magnetization process and the change of magnetization profile during the recovery process.

A theoretical model was developed to describe the formation of 3D heterogeneous magnetization profile. As shown in Figure R3.4, $\mathbf{M}'(x + \delta, y + u_y, z + u_z)$ and $\mathbf{M}(x, y, z)$ represent magnetization vectors under deformation and after recovery, respectively. Considering the recovery process, we can derive the relationship between \mathbf{M}' and \mathbf{M} , as shown in Eq. R3.19.

$$\mathbf{M}(x, y, z) = \frac{\mathbf{R}}{J} \mathbf{M}'(x + \delta, y + u_y, z + u_z) \quad (\text{R3.19})$$

$J = \det(\mathbf{F})$, where \mathbf{F} is the deformation gradient of an element of body shown in Figure R1.4. \mathbf{R} is the rotational component of \mathbf{F} . To estimate the magnetization vector after recovery, we focus on the rotation of the element along z axis and y axis with Eqs. R3.20-R3.22.

$$\mathbf{R} = \begin{bmatrix} \cos(\psi) & 0 & \sin(\psi) \\ 0 & 1 & 0 \\ -\sin(\psi) & 0 & \cos(\psi) \end{bmatrix} \times \begin{bmatrix} \cos(\theta) & -\sin(\theta) & 0 \\ \sin(\theta) & \cos(\theta) & 0 \\ 0 & 0 & 1 \end{bmatrix} \quad (\text{R3.20})$$

$$\theta = \arctan\left(\frac{\partial \delta}{\partial y}\right) = \arctan\left\{\frac{\pi A}{\lambda} [C_1 e^{-\alpha z} + C_2 e^{\alpha z} + C_3 \cos(\beta z) + C_4 \sin(\beta z)] \cos\left(\frac{2\pi}{\lambda} y\right)\right\} \quad (\text{R3.21})$$

$$\psi = \arctan\left(\frac{\partial \delta}{\partial z}\right) = \arctan\left\{\frac{A}{2} [-\alpha C_1 e^{-\alpha z} + \alpha C_2 e^{\alpha z} - \beta C_3 \sin(\beta z) + \beta C_4 \cos(\beta z)] \sin\left(\frac{2\pi}{\lambda} y\right)\right\} \quad (\text{R3.22})$$

There is a uniform pulse magnetic field applied to the buckled strip. Thus, $\mathbf{M}' = M_0 [1 \ 0 \ 0]^T$, where M_0 represents the magnitude of remanent magnetization. By combining Eqs. R3.19 and R3.20, we can obtain the expression of \mathbf{M} , as shown in Eq. R3.22.

$$\mathbf{M} = \begin{bmatrix} M_x \\ M_y \\ M_z \end{bmatrix} = M_0 \begin{bmatrix} \cos(\psi)\cos(\theta) \\ \sin(\theta) \\ -\sin(\psi)\cos(\theta) \end{bmatrix} \quad (\text{R3.23})$$

To validate the theoretical model, we employed a strip with $L = 28$ mm, $H = 2.7$ mm, $T = 0.5$ mm. The variation of magnetization vector along the y direction at $z=0$ and $z=H$ is shown in Figure R3.5(a). The magneto-optical result in Figure R3.5(b) demonstrates the distribution of the magnetic flux density which is consistent with the simulation result based on the calculated magnetization profile.

Figure R3.5. Magnetization profile of the strip and validation of magnetic field distribution.

a Magnetization profile of the strip at $z=0$ and $z=H$. **b** Validation of magnetic field distribution. The experimental result is obtained by a magneto-optical sensor and the modelling result is obtained based on the calculated magnetization profile.

(3) Magnetic response behavior:

Figure R3.6. Schematic diagram of the magnetic actuation process.

To elucidate the impact of external magnetic field on the deformation of the magnetic elastomer, a theoretical model to illustrate the relationship between magnetic torque and out-of-plane displacement (δ_m) was established. The out-of-plane displacement of the strip shown in Figure R3.6 can be calculated using plate theory for thin plate, as shown in Eqs. R3.24 and R3.25.

$$M_z = D \left(\frac{\partial^2 \delta_m}{\partial z^2} + \nu \frac{\partial^2 \delta_m}{\partial y^2} \right) = \iiint_V (-\tau_m) dV \quad (\text{R3.24})$$

$$\tau_m = [0 \quad 0 \quad 1] \mathbf{R}' \mathbf{M} \times \mathbf{B} \quad (\text{R3.25})$$

τ_m is the magnetic torque and M_z represents the bending moment applied to the elastomer. \mathbf{R}' is the rotational matrix for the change of magnetization direction during the deformation. To estimate the magnetic actuated deformation, we utilize small deflection assumption by approximating \mathbf{R}' as an identity matrix, and assume that the effective body torque distribution along the elastomer strip is equivalent to that generated by external bending moment applied at the free end of the strip [7. Zhao, Ruike, et al. "Mechanics of hard-magnetic soft materials." *Journal of the Mechanics and Physics of Solids* 124 (2019): 244-263.]. Furthermore, we express the out-of-plane displacement as $\delta_m(y, z) = \theta_m(y) \times \Lambda_m(z)$. With these simplifications, Eq. R3.26 can be obtained for the case that an external magnetic field is applied along the x direction with magnitude of B_0 .

$$D \left(\theta_m \frac{d^2 \Lambda_m}{dz^2} + \nu \Lambda_m \frac{d^2 \theta_m}{dy^2} \right) = \int_0^y \int_0^H \{ M_0 B_0 T \times \sin[\theta(y, z)] \} dy dz \quad (\text{R3.26})$$

The function $\sin[\theta(y, z)]$ can be fitted as: $\sin[\theta(y, z)] \approx G(z) \times \cos(ky + \phi)$. Based on the right side of Eq. R3.26, we can define θ_m as: $\theta_m = \sin(ky + \phi)$. Thus, Eq. R3.26 can be written as:

$$D \left(\frac{d^2 \Lambda_m}{dz^2} - \nu k^2 \Lambda_m \right) \sin(ky + \phi) = \int_0^H \left\{ \frac{M_0 B_0 T}{k} \times G(z) \right\} dz \times \sin(ky + \phi) \quad (\text{R3.27})$$

By solving Eq. R3.27 with the boundary condition ($\Lambda_m(z = 0) = 0$) numerically, we could obtain the result of $\delta_m = \Lambda_m \times \sin(ky + \phi)$. We employed a strip with $L = 28$ mm, $H = 4.3$ mm, $T = 0.5$ mm to validate the model, as shown in Figure R3.7. There is an agreement with the experimental result of the deformed strip.

Figure R3.7. Comparison of experimental results and theoretical model for the deformation of the strip under magnetic actuation.

In summary, we perform theoretical studies for the solvent and magnetic responsive behaviors of the magneto-elastomer. The theoretical models could give the quantitative relationship between the key parameters of buckling transformation and geometric parameters, the possible deformation modes of buckling structures, the quantitative description of 3D heterogeneous magnetization profiles, and the deformation under magnetic stimuli. The studies provide a

theoretical guideline for the structural design and understanding of the underlying transformation mechanisms.

2. Second, I wonder the buckled shape be critically dependent on the boundary conditions (Fig. 3). Is this true? This dependence is reproducible, i.e., the observed buckling shape was always observed under fixed experimental condition?? Boundary condition I, for example, the central part buckles into a complete wave (1 full wavelength), then the full wave can be either down-up or up-down, I think. This is because both the buckled shape seems to be in the same energy state, in terms of energy approach. This is true for other cellular structures.

Response: Thanks for your comment. Yes, as you stated, the buckling transformation is not deterministic. For these structures with different connectivity types, there will be either up-down or down-up waveforms, as shown in Figure R3.8(a) and (b). For lattice structures, the nodes would also exhibit different handedness. In addition, as the discussion in Figure R3.2, mixed buckling modes can be observed in the plate structure with critical value of L/H . For instance, when $L/H = 3.70$ (L and H are 10 and 2.7 mm, respectively), the number of wave exhibited by the buckled structure can be 1.0 or 1.5. Therefore, the buckling modes exhibited by soft structures are multi-stable [1. Bertoldi, Katia, et al. "Flexible mechanical metamaterials." *Nature Reviews Materials* 2.11 (2017): 17066; 2. Pal, Aniket, et al. "Exploiting mechanical instabilities in soft robotics: control, sensing, and actuation." *Advanced Materials* 33.19 (2021): 2006939.] and the final state after buckling process possess a certain randomness. The introduction of magnetic stimulation could help to predetermine the buckling states of soft architected materials, and enable the formation of definite and controllable buckling states without prolonged magnetic stimulation input. As shown in Figure R3.8(d)-(e), by applying magnetic stimulation first followed by the solvent stimulation, the structure can form different buckling states with the change of the direction of the external magnetic field. Such geometric transformation can be maintained even when the magnetic field is removed. The result indicates that external magnetic stimulation can help to determine the stable state of soft structure during buckling process, and moreover, this regulation does not require prolonged magnetic field stimulation input. According to your comment, we have added the discussion in "2.4 Morphological transformations of cellular structures" part of the revised manuscript and Figure R3.8 in the revised manuscript as Supplementary Fig. 9.

Figure R3.8. Multiple transformation results of strips with different connectivity types and lattice structures. **a-c** The dual transformation results of the elastomer structures; **d-e** Tuning the buckled state of square lattice via magnetic and solvent stimulation. The scale bars are 5 mm.

3. Regarding the selective particle trap, the experimental set-up seems not be correct: for opening of the cellular structure of 500 μ m, small (200 μ m) particle and large (600 μ m) particle mixture seems irrational. The authors should select the particle sizes just above/below the deformed size of cellular structure.

Response: Thanks for your kind suggestion. We are sorry for the confusion caused.

In the selective particle trapping and release experiments, our initial aim is to screen the particles smaller than the opening size of cellular structure. Under the actuation of a rotating permanent magnet, the surface topography of the cellular structure would periodically switch between open and closed states, which allows the reservation of particles smaller than the opening size (500 μ m) in chamber and the extrusion of particles larger than the opening size. If the sizes of all particles are smaller than the opening size (e.g., 200 μ m particles with green fluorescence and 400 μ m particles with red fluorescence), effective separation cannot be achieved as shown in Figure R3.9(a). In contrast, when the particle sizes are closer to the opening size of cellular structure, Figure R3.9(b) shows that 400 μ m particles (white color) are

reserved and 600 μm particles (yellow color) are removed, thus achieving selective trapping of 400 μm particles.

Moreover, according to your comment, additional experiment is provided here to demonstrate the separation of particles based on the deformed lattice size ($\sim 340 \mu\text{m}$), as shown in Figure R3.9(c). 200, 400 and 600 μm particles are used here. At first, dynamic transformation of cellular structure is triggered by a rotating magnet, resulting in the extrusion of 600 μm particles, and the reservation of 200 and 400 μm particles in the cellular chamber. Afterwards, a static magnetic field is imposed to deform the cellular structure, followed by the application of a flow disturbance. At this time, 400 μm particles are physically held by the boundary of cellular structure, while 200 μm particles are flushed away from the deformed gap of chamber. Therefore, this strategy can successfully screen out 400 μm particles from the group of 200, 400 and 600 μm particles, which is more effective and selective than the first method just based on the opening size of cellular structure. According to your comment, we have added the discussion and experimental results in the “**2.6 Potential applications of the dynamic transformation of non-freestanding structures**” part of the revised manuscript.

Figure R3.9. Schematic and demonstration of in-situ particle manipulation with the morphological transformation of the micro cellular structure. **a** Schematic illustration of initial and deformed states of the micro cellular structure and the unseparated microparticles. The micro cellular structure has an opening size of 500 μm , which allows the reservation of 200 μm (green) and 400 μm (red) particles in the chamber and the extrusion of 600 μm (yellow)

particles. By applying magnetic stimulus, the deformed cellular structure with a gap would be formed whose size can be regulated by the magnetic field strength. **b** All unseparated particles are smaller than the opening size (e.g., 200 μm particles and 400 μm particles), effective separation cannot be achieved through the dynamic structural transformation. **c** Selective particle trapping through the dynamic structural transformation. Particles of 400 (red) and 600 μm (yellow) in size are adopted. **d** Schematic illustration and demonstration of the selective trapping process using 200 (green), 400 (red) and 600 (yellow) μm particles. Dynamic transformation of cellular structure triggered by a rotating magnet results in the extrusion of 600 μm particles. Afterwards, a static magnetic field is imposed to deform the cellular structure with a gap of $\sim 340 \mu\text{m}$, followed by the application of a flow disturbance. At this time, 400 μm particles are physically held, while 200 μm particles are flushed away. The scale bars are 200 μm .

4. For fluidic flow generation experiments, the authors should give explicit power consumed to generate the flow, and discuss the result to other flow generation methods in terms of energy cost. Further, rather than those simple flow generation, they should show unique capability of their system, compared to typical methods. For example, I think the present flow generation may be much informative when it can be used to fluid mixing in microfluidic channel flow: this micro-scale fluid mixing is very costly, while one can generate kind of secondary flow using method proposed in this work to mix different fluids efficiently.

Response: Thanks for your kind suggestion.

The energy consumption and efficiency of fluidic manipulation is analyzed based on the proposed transformation in this work, which is verified as a relatively efficient fluidic manipulation method compared with those reported by other literatures. To further demonstrate the potential function of the method, we apply the proposed dynamic transformation to a microfluidic device, which can achieve efficient mixing of viscous liquids at a low Reynolds number environment as well as in a dynamic environment. According to your suggestion, we have added the following discussion in the “**2.5 Flow field induced by the morphological transformation under dynamic magnetic field**” part and “**2.6 Potential applications of the dynamic transformation of non-freestanding structures**” part of the revised manuscript.

(1) In this part, we calculate the energy consumption of the developed fluidic manipulation method, and perform the comparison of the pumping performance reported by other literatures.

To evaluate the pumping efficiency of fluid generation methods, a dimensionless parameter (ϵ_{ef}) similar to Ref. [1. Eloy, Christophe, and Eric Lauga. "Kinematics of the most efficient cilium." *Physical Review Letters* 109.3 (2012): 038101; 2. Osterman, Natan, and Andrej Vilfan. "Finding the ciliary beating pattern with optimal efficiency." *Proceedings of the National Academy of Sciences* 108.38 (2011): 15727-15732.] is adopted as Eq. R3.28.

$$\varepsilon_{ef} = \frac{\eta_{vis} Q^2}{V_{vol} P} \quad (R3.28)$$

η_{vis} and Q represent the viscosity of the pumping fluid and volume flow rate, respectively. V_{vol} is the volume of the magnetic elastomer used for pumping fluid and P is the input power. The volume flow rate $Q = 25.6 \mu\text{l}/\text{min}$ of the proposed method is obtained by measuring the change in mass of the fluid flowing from the outlet with the actuation frequency of 5 Hz [3. Zhou, Mingxing, et al. "Miniaturized soft centrifugal pumps with magnetic levitation for fluid handling." *Science Advances* 7.44 (2021): eabi7203.]. Glycerol is adopted as the fluid with a viscosity η_{vis} of $0.876 \text{ N} \cdot \text{s}/\text{m}^2$. The input energy P is determined by integrating the work performed by magnetic torque on the dynamic transformation of the magnetic elastomer [4. Ren, Ziyu, et al. "Multi-functional soft-bodied jellyfish-like swimming." *Nature Communications* 10.1 (2019): 2703.], as shown in Eq. R3.29, where $\boldsymbol{\omega}(x, y, z, t)$ represent angular velocity and T is the time for one actuation cycle.

$$P = \frac{1}{T} \int_0^T \int_{V_{vol}} |(\mathbf{R}' \mathbf{M} \times \mathbf{B}) \cdot \boldsymbol{\omega}| dV dt \quad (R3.29)$$

Eq. R3.30 is used to calculate the kinematic property of the magnetic elastomer over one actuation cycle.

$$\delta_m(x, y, z, t) = \Lambda_m(z) \times \sin\left(\frac{2\pi}{\lambda} y + \phi(t)\right) \quad (R3.30)$$

A_m is the magnitude of the out-of-plane displacement under magnetic actuation and $\phi(t)$ is the phase shift induced by the varying magnetic field direction and can be measured from experimental results. According to the analysis of magnetic response behavior, the rotational deflection and angular velocity can be obtained as Eqs. R3.31 and R3.32.

$$\alpha = \arctan\left(\frac{\partial \delta_m}{\partial y}\right) = \arctan\left\{\frac{2\pi \Lambda_m}{\lambda} \cos\left[\frac{2\pi}{\lambda} y + \phi(t)\right]\right\} \quad (R3.31)$$

$$\omega(x, y, z, t) = \frac{\partial \alpha}{\partial t} = \frac{-2\pi \Lambda_m}{\lambda \left[1 + \left(\frac{\partial \delta_m}{\partial y}\right)^2\right]} \cdot \sin\left(\frac{2\pi}{\lambda} y + \phi(t)\right) \cdot \frac{d\phi(t)}{dt} \quad (R3.32)$$

By applying a rotating magnetic field $\mathbf{B} = B_0 \left[\cos\left(\frac{2\pi t}{T}\right) \quad \sin\left(\frac{2\pi t}{T}\right) \quad 0 \right]^T$, the input energy can be calculated as Eq. R3.33.

$$P = \frac{1}{T} \int_0^T \int_{V_{vol}} \left| M_0 B_0 \cdot \sin\left(\frac{2\pi t}{T} - \alpha - \theta\right) \cdot \boldsymbol{\omega} \right| dV dt \quad (R3.33)$$

By solving Eq. R3.33 numerically, we could obtain the average input power as 1.4×10^{-3} W. The power done by magnetic torque contributes to the storage of strain energy, the variation of kinetic energy of the magnetic elastomer and fluid.

The comparison of the pumping efficiency ε_{ef} with other methods is shown in Table R3.1, which demonstrates that the developed flow generation method outperforms most reported soft pumping device in terms of pumping efficiency.

Table R3.1. Comparison of the pumping efficiency with literature

Refs	Actuation strategy	f (Hz)	Medium	Re	Q (ul/min)	P (W)	V (mm ³)	ε_{ef}
1	Magnetic actuation shape-memory pump	270	Glycerol solution	NA	1800	0.208	25.0	92.9
2	Magnetically actuated peristaltic motion	20	Water	NA	74.7	0.016	78.4	0.07
3	Chamber deformation via electromagnetic actuation	1~20	Water	NA	67.5	0.049	24.7	0.06
4	Cilium motion	31.8	Water	NA	2.1×10^{14}	2.5×10^{14}	5.4×10^{10}	30.4
5	Cilia array inspired motion	2.5	Glycerol solution	0.03	150	NA	NA	NA
6	Magnetically actuated centrifugal pump	1.6~16.6	Water and fat emulsion	NA	30600	4.1	NA	NA
This work	Magnetically actuated dynamic transformation	5	Glycerol solution	0.004	25.6	0.0014	25.9	262.6

Note: f : Actuation frequency; Q : Volume flow rate; P : Power; V : Volume of the actuator; ε_{ef} : Pumping efficiency (Dimensionless). [1. Smith, Aaron R., et al. "Characterization of a high-resolution solid-state micropump that can be integrated into microfluidic systems." *Microfluidics and Nanofluidics* 18.5 (2015): 1255-1263; 2. Saren, A., A. R. Smith, and K. Ullakko. "Integratable magnetic shape memory micropump for high-pressure, precision microfluidic applications." *Microfluidics and Nanofluidics* 22.4 (2018): 38; 3. Rusli, M. Q. A., et al. "Electromagnetic actuation dual-chamber bidirectional flow micropump." *Sensors*

and Actuators A: Physical 282 (2018): 17-27; 4. Eloy, Christophe, and Eric Lauga. "Kinematics of the most efficient cilium." *Physical Review Letters* 109.3 (2012): 038101; 5. Dong, Xiaoguang, et al. "Bioinspired cilia arrays with programmable nonreciprocal motion and metachronal coordination." *Science Advances* 6.45 (2020): eabc9323; 6. Zhou, Mingxing, et al. "Miniaturized soft centrifugal pumps with magnetic levitation for fluid handling." *Science Advances* 7.44 (2021): eabi7203.]

(2) In this part, the dynamic transformation of buckling structure is used for efficient mixing of viscous fluids in microfluidic devices at low Reynolds number.

Figure R3.9. Fluid mixing at low Re using the dynamic transformation of the magneto-elastomers. **a** Schematic diagram of the microfluidic device with Y-shaped fluidic channel. **b** Images of the fluid mixing device including glycerol layers with two colors and magneto-elastomers. **c** Dynamic transformation of the magneto-elastomers in the fluidic channel under rotating magnetic field. **d** Efficient fluid mixing with 0 flow rate of the inlet. **(e)** Efficient fluid mixing with 40 mL/h flow rate of the inlet. The scale bars are 3 mm.

The mixing of fluids in small-scale channels is important for the development and application of microfluidic devices. Nevertheless, efficient mixing is challenging due to the low Reynolds number (Re) and slow diffusion effects. Existing methods include passive fluid mixing devices, which often require specific channel designs, such as spiral microchannels, or straight channels with groove or ridge microstructures [1. Stroock, Abraham D., et al. "Chaotic mixer for microchannels." *Science* 295.5555 (2002): 647-651; 2. Therriault, Daniel, Scott R. White, and Jennifer A. Lewis. "Chaotic mixing in three-dimensional microvascular networks fabricated by direct-write assembly." *Nature Materials* 2.4 (2003): 265-271; 3. Zhao, Qianbin, et al. "A review of secondary flow in inertial microfluidics." *Micromachines* 11.5 (2020): 461.]. In addition, active mixing devices have also been developed, including magnetically actuated artificial cilia and bubble-based microfluidic devices, which may have limited mixing

thoroughness or need special design of ciliary arrays to achieve optimized metachronal wave [4. Dong, Xiaoguang, et al. "Bioinspired cilia arrays with programmable nonreciprocal motion and metachronal coordination." *Science Advances* 6.45 (2020): eabc9323; 5. Khoshmanesh, Khashayar, et al. "A multi-functional bubble-based microfluidic system." *Scientific Reports* 5.1 (2015): 9942.]. Here, we apply magneto-elastomers with dynamic structural transformations for fluid mixing at low Re in microfluidic devices, as shown in Figure R3.9. Glycerol colored with different dyes is used as the fluid at the inlet and fills the entire microfluidic channel. The volume of the straight channel is 182 mm^3 , and no apparent diffusion of fluid occurs in the channel under static state. Two magnetic strips with a height of 1.6 mm are used for the fluid mixing. With a dynamic magnetic field (actuation frequency: 5 Hz), we demonstrate that this mixing device could rapidly mix viscous glycerol with local Re less than 0.004. In addition, we also verify the efficient fluid mixing under external dynamic fluid environment (flow rate: 40 mL/h).

Overall, I think the manuscript can be published once the authors revise their manuscript properly.

Response: Thanks a lot for your comment. We have carefully revised our work according to your all suggestions. We hope such revision can address your questions at a satisfactory level and convince you of the suitability of our work for publication in *Nature Communications*.

List of Changes in Manuscript

1. “Jinsheng Zhao” and “Liu Wang” are added in the author list.
2. “topological transformation” is changed to “morphological transformation” or “transformation”.
3. “boundary condition” is changed to “connectivity type”.
4. In **abstract**, “The introduction of magnetic stimulation could help to predetermine the buckling states of soft architected materials, and enable the formation of definite and controllable buckling states without prolonged magnetic stimulation input.” is added.
5. In **abstract**, “The dynamic modulations can be exploited to build systems with switchable fluidic properties and are demonstrated to achieve capabilities of fluidic manipulation, selective particle trapping, and sensitivity-enhanced biomedical analysis.” is changed into “The dynamic modulations can be exploited to build systems with switchable fluidic properties and are demonstrated to achieve capabilities of fluidic manipulation, selective particle trapping, sensitivity-enhanced biomedical analysis, and soft robotics.”.
6. In paragraph 1 of **Introduction**, “The active topological transformation (e.g., buckling or wrinkling behaviors) of three-dimensional (3D) morphable structures has attracted great attention due to their geometry-determined functionalities.” is changed into “Three-dimensional (3D) morphable structures with the active transformation capability have attracted great attention due to their geometry-determined functionalities and dynamic interactions with environment.”.
7. In paragraph 2 of **Introduction**, “topological structures” is changed into “soft structures”; “topological defects” is changed into “artificial defects”; “topological transformation behaviors” is changed into “morphological transformation behaviors”.
8. In paragraph 3 of **Introduction**, “topological structures” is changed into “soft morphable structures”; “Some previous studies focused on mechanical or metamaterial systems constructed by magneto-elastomers with relatively uniform magnetization profiles, which may limit the degree of deformation freedom and the corresponding functionalities” is added; “achieving the buckling or wrinkling behaviors in nature or other complex topological transformations” is changed into “achieving the buckling, wrinkling or other complex morphological transformation behaviors”.
9. In paragraph 1 of **2.1 Shape-morphing mechanisms of the magneto-elastomers**, “When the non-freestanding structure is immersed in the organic solvent, the difference in the volume expansion rate along the height direction could induce the geometric transformation (the buckled state in Fig. 1) of the structure. The elastomeric structure near the substrate will not swell due to the constraint of the substrate, while the side away from the substrate will swell by absorbing the organic solvent.” is changed into “When the magneto-elastomer is immersed in toluene solution, swelling behavior of elastomer structure is induced by the diffusion of toluene into polymer matrix. While the swelling process of bottom part would be affected by the constraint of substrate, axial compressive forces are formed to provide the buckling condition of the swelled elastomer (the buckled state in Fig. 1).”.

10. “Characterization of magnetic-responsive behavior of surface-attached structures.” is changed into “**2.2 Characterization of magnetic and solvent responsive behavior of surface-attached structures**”.
11. In paragraph 2 of **2.2 Characterization of magnetic and solvent responsive behavior of surface-attached structures**, “The shape-morphing results are also affected by the magnetic field strength and magnetic field direction.” is added; “The impact of magnetic field strength and direction angle on the wavelength is shown in Supplementary Fig. 6, where both magnetic field density and direction angle have no effect on the wavelength.” is added.
12. Paragraph 3 of **2.2 Characterization of magnetic and solvent responsive behavior of surface-attached structures** that discuss relevant analytical models is added.
13. **2.3 Geometric transformation of magneto-elastomer under the coupling stimulation** is added.
14. Paragraph 3 of **2.2 Characterization of magnetic-responsive behavior of surface-attached structures** is changed into paragraph 1 of **2.4 Morphological transformations of cellular structures**.
15. In paragraph 1 of **2.4 Morphological transformations of cellular structures**, “It should be noted that the buckling transformations for these structures with different connectivity types are also not deterministic, as there will be either up-down or down-up waveforms, as shown in Supplementary Fig. 9. Further study of connections with different constraints of degrees of freedom (including translational and rotational degree of freedom) is performed via FEA models, as shown in Supplementary Fig. 10. The two sides of the plate structure distributed with non-responsive material and fixed constraints would induce a bell-shaped deformation under solvent stimulation which is different from that with free-to-rotate junctions.” is added.
16. Paragraph 2 of **2.4 Morphological transformations of cellular structures** is added.
17. In paragraph 3 of **2.4 Morphological transformations of cellular structures**, “In addition, the impact of the coupled stimulation on the deformation of lattice structure is also studied, as shown in Supplementary Fig. 13 and 9, which are consist with the results of aforementioned strip structure, demonstrating the capability of regulating the buckling states of lattice structures.” is added.
18. In paragraph 2 of **2.5 Flow field induced by the morphological transformation under dynamic magnetic field**, “In addition, we calculate the energy consumption of the developed fluidic manipulation method, and perform the comparison of the pumping performance reported by other literatures, as shown in Supplementary Note 2 and Supplementary Table 1, which demonstrates that the developed flow generation method outperforms most reported soft pumping device in terms of pumping efficiency.” is added.
19. In paragraph 1 of **2.6 Potential applications of the dynamic transformation of non-freestanding structures**, “and three kinds of particles are used with the size of 200, 400 and 600 μm (Supplementary Movie 7). At first, dynamic transformation of cellular structure is triggered by a rotating magnet, resulting in the extrusion of 600 μm particles, and the reservation of 200 and 400 μm particles in the cellular chamber. Afterwards, a static

magnetic field is imposed to deform the cellular structure with the formation of a $\sim 340 \mu\text{m}$ gap, followed by the application of a flow disturbance. At this time, $400 \mu\text{m}$ particles are physically held by the boundary of cellular structure, while $200 \mu\text{m}$ particles are flushed away from the deformed gap of chamber. Therefore, this strategy can successfully screen out $400 \mu\text{m}$ particles from the group of 200 , 400 and $600 \mu\text{m}$ particles. In addition, Supplementary Fig. 21 and Supplementary Movie 7 show the separation of microparticles by only using the dynamic transformation of the microcellular structure, which allows the reservation of particles smaller than the opening size ($500 \mu\text{m}$) in chamber. For instance, $200 \mu\text{m}$ or $400 \mu\text{m}$ particles can be selectively trapped.” is added.

20. Paragraph 3 and 4 of **2.6 Potential applications of the dynamic transformation of non-freestanding structures** that discuss the application of efficient fluid mixing and untethered swimming robots are added.
21. In **Conclusions**, “Using buckling instability to encode the internal magnetization profiles of the magneto-elastic structure enables it to generate spatially distributed magnetic torques under magnetic stimuli, thereby driving the topological transformation of architected materials. Programmable magnetic field inputs including strengths, directions, and gradients facilitate the realization of multimodal anisotropic topological transformations which well beyond their quasi-static states.” is changed into “A facile magnetization programming method by using the buckling instability phenomenon is proposed, which provides a template-free and on-demand manner for the formation of magnetoactive materials with 3D magnetization profile. Through programming magnetic field inputs including magnetic field direction, strength, and gradient, the buckling-induced magnetized elastomers could exhibit multi-mode anisotropic transformations or robust switch between different buckling states.”; “efficient fluidic mixing in microfluidic device, and untethered swimming robot.” is added.
22. In Materials of **Experimental Section**, “rhodamine B, methylene blue, crystal violet, isobornyl acrylate.” is added.
23. In **Experimental Section**, “Preparation of magneto slurry” is added.
24. In Fabrication of 3D non-freestanding structures of **Experimental Section**, “the PDMS mold was treated with plasma for 10 min following with silane steam treatment for 3 h using triethoxy(1H,1H,2H,2H-perfluoro-1-octyl)silane (POTS). After the silane treatment, the PDMS surface was cleaned via ethanol and dried in the oven ($60 \text{ }^\circ\text{C}$) for 1 h. The magnetic slurry was poured onto the surface of the PDMS mold, and were placed in a vacuum chamber for 5 min so that the slurry could thoroughly fill the cavity portion. A glass substrate with hydrophilic surface obtained by plasma treatment for 10 min was covered on top of the magnetic slurry which was cured in the oven ($60 \text{ }^\circ\text{C}$) for 20 min. The cellular elastomer structure was released from the acrylic mold and stayed on the glass substrate.” is added; “and poly(ethylene glycol) diacrylate (PEGDA) resin. To avoid the residual resin affecting the curing of the elastomer, the resin mold was irradiated by UV light for 1 hour, and cleaned via ethanol, and followed by the hydrophobic treatment. The magnetic slurry fills the microgrooves of the resin mold in a vacuum chamber and is

covered by a hydrophilic-treated glass substrate. After the magnetic elastomer was cured, the cellular microstructure was peeled off from the resin mold.” is added.

25. In PIV analysis of **Experimental Section**, “The distance between the air-water interface and the upper surface of the elastomer structure was kept at 3 mm. Hirox RH-2000 Digital Microscope was used to record the process.” is added.
26. In **Experimental Section**, “**Droplet manipulation via the transformation of the strip structures**”, “**In-situ particle manipulation**”, “**Collection of sprayed aerosol**”, “**Fabrication and actuation of untethered swimming robots**”, “**Fluidic mixing device**” are added.
27. In Simulation methods of **Experimental Section**, “(iv) The simulation of the coupling stimulation of magnetic field and solvent was performed by COMSOL. Elastomer structures were divided into smaller subsections with varying magnetic torques in COMSOL structural mechanics module. The torques of subsections were recalculated according to the deformation of the elastomer structure until it reaches the equilibrium state. Subsequently, solvent stimulation was applied. The deformed elastomer structure could undergo the swelling process and buckling transformation which are simulated through thermal expansion function.” is added.
28. In Characterization Techniques of **Experimental Section**, “The magnetic hysteresis of the magnetic elastomer was measured by a PPMS model 6000 Quantum Design VSM.” is added.
29. In Author contribution of **Acknowledgments**, “N. X., L. W., and J. Z., performed the simulation and theoretical studies for the active deformation under stimuli from solvent and magnetic field.” is added.
30. Figure 1, Figure 2, and Figure 7 are revised.
31. Figure 3 and Figure 4 are added.
32. New references 50, 51, 61, 62, 63, 64 are added.

List of Changes in Supporting Information

1. Supplementary Note 1 Analytical model is added.
2. Supplementary Note 2 Pumping efficiency is added.
3. Supplementary Note 3 Magnetic field generation for the actuation of the untethered swimming robot is added.
4. New Supplementary Figures 6, 9, 10, 11, 13, 15, 21, 22, 23, 24 are added.
5. New Supplementary Table 1 is added.
6. New Supplementary Movies 2, 9, 10 are added.
7. Demonstration of different separation methods are added in revised Supplementary Movie S7.
8. New references 1-11 are added.

REVIEWERS' COMMENTS

Reviewer #1 (Remarks to the Author):

In the revised manuscript, authors included FEA simulation that couples solvent and magnetic field, which increases the novelty of the work. In addition, new applications in soft robotics were added, which will increase the impact of the work. In addition, the authors address all my concerns and comments that i raised in the previous review process. Thus, i suggest it for publication.

Reviewer #2 (Remarks to the Author):

The authors have addressed all my questions satisfactorily. The analytical studies are thorough and agree well with the experiments. The new demonstrations for potential applications are interesting and showcase the diversity of possible use.

Reviewer #3 (Remarks to the Author):

The authors have revised their manuscript according to the points raised by the reviewers, and thus I think it can be published in Nat. Comm.

Response to Reviewer #1

In the revised manuscript, authors included FEA simulation that couples solvent and magnetic field, which increases the novelty of the work. In addition, new applications in soft robotics were added, which will increase the impact of the work. In addition, the authors address all my concerns and comments that i raised in the previous review process. Thus, i suggest it for publication.

Response: We sincerely thank the reviewer for the kind comments and helping us improve the quality.

Response to Reviewer #2

The authors have addressed all my questions satisfactorily. The analytical studies are thorough and agree well with the experiments. The new demonstrations for potential applications are interesting and showcase the diversity of possible use.

Response: We sincerely thank the reviewer for the kind comments and helping us improve the quality.

Response to Reviewer #3

The authors have revised their manuscript according to the points raised by the reviewers, and thus I think it can be published in Nat. Comm.

Response: We sincerely thank the reviewer for the kind comments and helping us improve the quality.